# Experimental Study on Angular Flexural Performance of Multiaxis Three Dimensional (3D) Polymeric Carbon Fiber/Cementitious Concretes

**DOI:** 10.3390/polym13183073

**Published:** 2021-09-11

**Authors:** Huseyin Ozdemir, Kadir Bilisik

**Affiliations:** 1Vocational School of Technical Sciences, Gaziantep University, Gaziantep 27310, Turkey; hozdemir@gantep.edu.tr; 2Nano/Micro Fiber Preform Design and Composite Laboratory, Department of Textile Engineering, Faculty of Engineering, Erciyes University, Talas-Kayseri 38039, Turkey; 3Nanotechnology Application and Research Centre (ERNAM), Erciyes University, Talas-Kayseri 38039, Turkey

**Keywords:** carbon fiber, multiaxis preform concrete, angular flexure, fracture toughness, failure mode

## Abstract

Multiaxis three-dimensional (3D) continuous polymeric carbon fiber/cementitious concretes were introduced. Their angular (off-axis) flexural properties were experimentally studied. It was found that the placement of the continuous carbon fibers and their in-plane angular orientations in the pristine concrete noticeably influenced the angular flexural strength and the energy absorption behavior of the multiaxis 3D concrete composite. The off-axis flexural strength of the uniaxial (C-1D-(0°)), biaxial (C-2D-(0°), and C-2D-(90°)), and multiaxial (C-4D-(0°), C-4D-(+45°) and C-4D-(−45°)) concrete composites were outstandingly higher (from 36.84 to 272.43%) than the neat concrete. Their energy absorption capacities were superior compared to the neat concrete. Fractured four directional polymeric carbon fiber/cementitious matrix concretes limited brittle matrix failure and a broom-like fracture phenomenon on the filament bundles, filament-matrix debonding and splitting, and minor filament entanglement. Multiaxis 3D polymeric carbon fiber concrete, especially the C-4D structure, controlled the crack phenomena and was considered a damage-tolerant material compared to the neat concrete.

## 1. Introduction

Textile concrete composite (TCC) consists of two-dimensional (2D) biaxial fabric or three-dimensional (3D) preform with multiple continuous fibers as the reinforcement materials and fine-grained cement as the matrix material. TCC can be considered noncorrosive, with better fracture toughness probably due to crack control via unusual stress transfer mechanism and lightweight compared to the steel concretes. Research on the TCC especially two-dimensional (2D) fabric reinforced concretes, was reported by several well-known researchers [1,2,3,4,5]. Fiber-reinforced concrete (FRC), on the other hand, elementally consists of staple fibers and cementitious matrix [6]. Textile concrete composites generally encompass high strength and high modulus fibers, including polyethylene, para-aramid, AR-glass, basalt, and carbon.

Textile technologies were used to make various 2D and 3D structures to apply in concrete end-uses [7,8]. Non-interlaced or interlaced large or small mesh size fabrics, oblique fabric, 2D warp knitted fabric, and 3D multiaxis warp knit fabrics are all single layer or layered substrates for concrete [7]. Additionally, multiaxis-layered substrate, orthogonal fabric, including spacer fabric with through-the-thickness fibers, are 3D preforms and they have been employed in concrete composites [9,10]. Hand lay-up, prestressed, pultrusion, spraying, and filament winding are some of the concrete manufacturing technology widely used for textile concrete composites [11,12,13]. Recently, it was reported that additive manufacturing (3-D printing) was used to make structural parts via layer-to-layer addition for various applications from medical to civil engineering [14,15]. Some of the possible application areas of textile concrete composites are prefabricated panels, industrial floors, bridge decks, and rehabilitations of aged concretes (connector, anchoring, confinement, jackets, wrapping, and seismic retrofit) [16,17,18]. Fiber tensile strength and modulus, fabric architecture and fiber angular orientations, fiber volume fraction, filament TOW (an untwisted bundle of continuous filaments) linear density, filament diameter, matrix properties including grain size and interlaminar bonding between filamentary TOW, and cementitious matrix are of paramount importance during engineering design, analysis, and characterizations of the textile concrete composites [19,20]. Lately, the interfacial properties in fiber-based concrete were enhanced by using nano particles such as discrete nanospheres, nanotubes, nanoplate, and staples nanofibers [21,22,23].

It has been observed that concrete beams with bars made of carbon fiber-reinforced polymer (FRP) exhibited worse resistance to cracking and faster crack propagation during the entire flexural loading process than steel bars. Flexural shear failure was observed due to the cracking patterns in FRP-based concrete bars [24]. However, strong bond strength was achieved in the carbon rod/cementitious concrete compared to the steel rod concrete. The hydrates in contact with carbon fiber in the interface region of concrete were the same as in the steel rod [25]. Peled and Bentur claimed that the fabric weave pattern in the flexural properties of concrete enhanced the bonding and strain hardening properties by using low modulus polyethylene and polypropylene yarns [26,27]. Another study showed that fine-grained concrete reinforced with carbon fabric with high yarn density increased the flexural strength, and the influence of the weave design was not significant [28]. On the contrary, recycled glass and carbon fiber reinforced concrete have slightly increased their flexural strength [29]. The recycled staple carbon fiber addition in concrete increased its flexural strength at the expense of capillary water absorption and electrical resistivity [30,31,32].

Geopolymer cement, which was alkaline-based, including fly ash or slag from iron and metal, and carbon fiber bars concrete demonstrated low crack width than the ordinary Portland cement-based concrete [33]. Biaxial carbon fabric with strain hardening cements matrix concrete exhibited premature local failure resulting in ductile behavior due to concrete softening mechanism [34]. Carbon fabric with staple steel and polymer fibers raised the flexural capacity, stiffness, crack initiation, procreation, and ductility of the concrete beams [35]. On the contrary, a single-layer carbon fabric reinforced concrete with short basalt fibers improved the flexural properties of concrete due to the fiber volume ratio and the strong bond strength between the fiber and cement matrix [36].

Carbon and glass fabric reinforced mortar with micro steel fiber was reported to enhance the energy dissipation capability of concrete panels, but their flexural properties were not improved [37]. The addition of steel fiber under prestressed carbon fabric concrete increased the flexural behavior of the TCC due to better interfacial bonding of the textile layers [38]. It was found that fatigue strength and energy absorption of the textile concrete composite were improved by the fiber strengthening phenomena [39]. It was realized that fine filament fibers in textile-reinforced concrete composites have a higher load-bearing performance than coarse filament fibers because more fibers in the fine fiber TOWs were in direct contact with the matrix than the coarse fiber TOWs. This eventually affected the filament-cement bond strength [40,41]. Other research has demonstrated that the interfacial bonding between small mesh size prepreg fabric and sand was enhanced due to the multiple cracks and their distributions. Therefore, fabric mesh affected the textile concrete composites’ bearing capacity and first-crack load [42].

The 3D glass woven preform was interlaced to form a thick substrate incorporating polyurethane/epoxy and cementitious matrix for making textile concrete composite [43]. The flexural properties of the 3D spacer reinforced cement composite were affected by Z-fiber density, 3D fabric architecture, and directional fiber volume fractions [44]. Another study showed that using the high-strength and high-modulus fiber TOWs in the concrete improves the performance of the panel strength and the absorbed toughness energy [45,46,47]. Feature foam-based concrete with calcium carbonate whisker and staple polypropylene fiber increased fracture performance under four-point bending loading due to a synergistic reinforcing effect [48]. Carbon fiber reinforced polymer (CFRP) retrofitted reinforced concrete (RC) delayed crack propagation and resisted beam deformations [49]. Concrete reinforced with 2D fiber-based fabric was influenced by fabric architecture, fiber volume fraction, and no effect was found in the span-to-depth ratio during three-point bending [50].

In the case of textile reinforced mortar (TRM) jackets, the U-wrapping strengthening configuration is much more effective than the side-bonding configuration, and increasing the number of layers significantly increases the effectiveness of the TRM jacket [51]. Another study reported that near-surface positioned U-jacketed carbon fabric concrete improved the mode-I delamination failure compared to the neat concrete [52]. It was demonstrated that anchoring the carbon fabric/epoxy composite to the concrete surface improved its ductility and crack arresting mechanism compared to the adhesively joint structures [53]. Concrete surface bonded carbon fabric composite externally bonded to the concrete with cement-based adhesive showed better flexural strengthening properties compared to the epoxy adhesive [54]. Carbon fabric composite was used as a stirrup element in concrete structures to enhance the shear properties. Furthermore, it was found that failure mode depended on the shear span/depth ratio and illustrated diagonal cracks [55]. The fracture toughness of staple high strength and modulus fiber reinforced concrete increased due to complex fiber-cement crack opening resistance and crack propagations [56]. The prevailing failures of short carbon fiber added reinforced concrete were fiber slippage and tensile fiber breakages in the cementitious matrix [57]. It was illustrated that the mechanical performances of impregnated carbon fiber concrete at elevated temperatures were improved due to using reactive mineral suspensions on the carbon fiber surface [58]. It was found that surface-functionalized staple carbon fiber demonstrated homogeneous dispersion and strong bonding in the cementitious matrix [59]. It was illustrated that tensile strength and modulus of polypropylene/cement concrete were raised due to the addition of nanoclays. Furthermore, the electromagnetic interference (EMI) shielding effectiveness and mechanical properties of concrete were improved with the incorporation of carbon nanotubes (0–2.0 wt.%), probably due to the denser calcium silicate hydrate (or C-S-H) structure, as well as nano bridging and pore-filling [60,61]. The porosity of ultra-high-strength concrete was reduced by about 30% by adding carbon nanofibers (CNFs, 0.3%), which influenced the internal morphology [62]. Further, it was reported that the addition of carbon nanofibers and foam steel/cementitious lightweight concrete have comparable mechanical and bond strength with ordinary steel concrete [63].

The fracture toughness of the cementitious matrix was increased by debonding fiber-matrix interfaces, fiber entanglement, pulling out, and stick-slip phenomena. Thus, fibers in the concrete structure administered the micro/macro crack propagation [64,65]. The fiber-cementitious matrix debonding, bridging, and pull-out affected energy dissipation during the fracture process in which the slow, stable crack-width growth probably improved the fiber concrete toughness [66]. Staple fiber/cement concrete showed tension-stiffening characteristics [67,68,69,70,71]. Naamen described that a “strain-hardening” phenomenon occurred in the high strength/modulus fiber concrete during tensile loading in that stress increased with the strain after the first crack formation. After that, multiple cracking appeared up to the maximum post-cracking stress before the “strain-softening” stage [72,73]. It was also claimed that the incorporation of the continuous or short fiber into the cementitious matrix improved the strength properties of the concrete due to the intricate stress transfer phenomena in which the energy absorption mechanism is via the fiber-matrix interaction as a form of debonding, bridging, and pull-out processes [74,75,76].

The artificial neural network (ANN) method was used to model the flexural strength of TCC where concrete with carbon fiber fabric and short fibers was used to improve the initial crack and post crack load capacity [77]. Carbon fibers in the textile-reinforced mortar (TRM) affected its failure modes and the shear behavior, which was verified by a crack-shear slip model [78]. Textile concrete composite (TCC) was modeled using a discrete approach, whereby textiles and matrix were considered separately, or the mechanical responses of the material were averaged over the cross-section [44]. Crack growth in short fiber concrete was analyzed with regard to crack instability and crack propagation conditions. Fiber aspect ratio, interfacial friction, and interfacial shear strength were considered critical parameters [79]. It was found that the pulling out of fibers (fiber toughening) and the complex fiber/matrix aggregate responses in steel fiber concrete were the main mechanisms during the strain hardening stage [80].

Numerous research studies have been carried out on concrete with added staple fiber or fabric substrate. However, limited research on the mechanical properties of the 3D concrete structure was examined considering a dry, continuous, filamentous TOW-based layered architecture with large meshes. Thus, the research aimed to develop multiaxially oriented high-strength and high modulus continuous carbon fiber-based concrete composite and to search experimentally the principal and angular (off-axis) flexural properties of these concrete composites.

## 2. Materials and Methods

### 2.1. Multiaxis 3D Preforms and Concretes

Polyacrylonitrile (PAN) based carbon fiber (AKSACA, DowAksa, Yalova, Turkey) were employed to design the multiaxis three-dimensional (3D) preforms. The properties of the carbon fibers are exhibited in Table 1. The tensile strength, modulus, and elongation at break of PAN-based carbon fibers were 4200 MPa, 240 GPa, and 1.80%, respectively, as provided in Table 1. Moreover, the PAN-based carbon fiber’s linear density, density, and diameter were 1600 tex (24 K, sizing type: A-42, amount: 0.5–1.0 wt.%), 1.76 g/cm^3^, and 6 μm, respectively.

A cementitious matrix including cement (Portland CEM I 52.5, Limak, Gaziantep, Turkey), siliceous fine sand (SiO_2_ based silica, according to AS 114131, fine aggregate 100–400 μm, density 2.65 g/cm^3^, melting temperature 1750 °C, Pomza export, Manisa, Turkey), superplasticizer (based on polycarboxylate according to ASTM C494 type F, GLENIUM^®^ 51, BASF, Ludwigshafen, Germany) to enhance the workability and rheology, and water (city water) was used to produce the multiaxis three-dimensional (3D) continuous carbon fiber reinforced concretes.

Multiaxis three-dimensional (3D) preform structures were chosen for the study due to eliminating interlacement-based crimp exchange and crimp extension (or displacement), as in the case of biaxial fabric [82]. This probably causes the stress concentration and leads to an ineffective stress transfer in the interlacement region of the fiber TOW around the biaxial fabric local region where the strength performance and fracture mechanism are probably adversely affected [83,84,85]. In addition, the layered (four layer) structure was made sparsely, and two of them were arranged at an angle inside the panel to homogenize the load-bearing capacity of the concrete. The fiber TOW in the multiaxis 3D preform and concrete was continuous form considering continuity principle. A specially developed flat winding-molding rig was used to make the multiaxis 3D preform, in which the preform structures were sparsely formed under constant tension (approximately 4 kg) by winding the four fiber sets as exhibited in Figure 1a–c [45,46]. The wooden laminate was employed to design and construct the winding apparatus. Its dimensions were 40 (length) cm × 40 (width) cm × 3 (thickness) cm. The fiber TOWs were positioned under pre-tension (4 kg) via precisely drilled holes in the in-plane and through-the-thickness directions. The mold surface was made using the lubricative layer to prevent fiber breakages during winding and easy demolding after casting.

In the flat winding/molding rig, the fibers, which were plied under low twists for increasing the volume fraction and prevent spreading the filaments, were multiaxially, biaxially, uniaxially oriented without any interlacements. As indicated in Table 2, the twist and areal density of the preforms are 10 turn/m and 10 ends/15 cm, respectively. Furthermore, the areal mesh size for C-1D, C-2D, and C-4D was 15 mm, 15 × 15 mm, and 15 × 15 × 15 × 15 mm, respectively. The layer-to-layer through-the-thickness distance was 6 mm. It was observed that the flat winding was a simple and quick process for manufacturing the multiaxis multilayered preforms for concrete panels. In the future, multiaxis 3D windings can be combined with 3D printing to produce one-piece concrete panels for various applications. For this reason, 3D printing and multiaxis 3D molding method can be merged, such as integrated the multifunctional robotic winding hand and cement mixture and casting via three-dimensional printing.

All the multiaxis 3D carbon concrete structures (C-1D, C-2D, and C-4D) was principally composed of four layers with different directions such as uniaxial (0°), biaxial (0°/90°), and multiaxial (90°/±45°/0°) direction. Some of the multiaxis 3D carbon concrete composites are shown in Figure 2a–c. As provided in Table 2, the stacking sequences of C-1D, C-2D, and C-4D were [0°]_4_, [90°/0°]_2_, and [90°/±45°/0°]_1_, respectively.

The PAN carbon filamentary TOWs preforms were consolidated via hand lay-up to make multiaxis 3D concrete composites. The multiaxis 3D concrete composite processing steps are shown schematically in Figure 3. A Hobart-type mixer (20-L capacity) was used to prepare the cementitious matrix mixtures. First, the solid constituents such as Portland cement (735.84 kg/m^3^, 34.51, wt.%) and siliceous fine sand (1030.35 kg/m^3^; 48.33, wt.%) were weighed and then mixed at 100 rpm for a minute (Figure 3). After that, water (364.91 kg/m^3^; 17.12, wt.%) and superplasticizer (0.917 kg/m^3^; 0.04, wt.%) were added with the dry mixture and mixed for one minute at 150 rpm [86]. Then a homogeneous cementitious matrix was achieved by mixing again at 300 rpm for two minutes. Considering the preform architecture design, PAN-based carbon TOWs were wound using the specially developed flat winding-molding rig (Figure 1). The cementitious mixture was introduced to the flat winding rig using the hand lay-up technique (Figure 3). The flat winding rig was manually shaken during the matrix inclusion to obtain complete penetration of the cementitious matrix into the opening of the preform and to contribute a strong preform-matrix bonding. After casting at atmospheric conditions (65 ± 2% relative humidity and 22 ± 2 °C), matrix inserted winding-molding rig was left 24 h for obtaining rigid consolidation. Then, all panel samples in the winding rig were removed. Finally, all multiaxis 3D carbon fiber concrete samples were wrapped with polyethylene film (Reverans Plastik Ambalaj Ltd., Ankara, Turkey) to cure at 95 ± 5% relative humidity and 23 ± 2 °C for 28 days (Figure 3), and samples were ready for the bending test.

### 2.2. Flexural Test for Multiaxis 3D PAN Carbon Concrete

A flexural test using four-point loading was conducted to determine the flexural properties of the multiaxis 3D carbon concrete composite panels. ASTM C78/C78M-18 [87] and ASTM C651-15 [88] standards were generally considered for the flexural test, but they were not exactly followed due to structural differences between the staple fiber reinforced concrete, and the multiaxis 3D preform concretes. The actual and schematic images of the flexural test are portrayed in Figure 4a–f. Measured sample sizes of the flexural test were 400 mm (length) × 75 mm (width) × 30 mm (thickness). The angular (off-axis) flexural test on the C-1D, C-2D, and C-4D samples was conducted in the principal direction (0° and 90°) and ±bias directions (+45° and −45°). The dimensions of the principal and off-axis flexural samples were drawn on the top surface of the multiaxis 3D fiber concrete panel. They were cut by using the wet concrete cutting machine, which has a diamond-tipped saw. Furthermore, the prepared samples were coded based on test directions, as C-1D-(0°), C-1D-(90°), C-1D-(+45°), C-2D-(0°), C-2D-(90°), C-2D-(+45°), C-4D-(0°), C-4D-(90°), C-4D-(+45°), and C-4D-(−45°). The flexural test data were obtained using a Shimadzu AG-X 100 (JP) testing instrument accompanied by Trapezium^®^ software with a 100 kN loading cell. The cross-head speed was 0.9 mm/min. The flexural tests in the principal and ±bias directions were repeated 5 times and 3 times, respectively. The flexural test was done at the standard laboratory atmosphere condition (temperature of 23 ± 2 °C and relative humidity of 50 ± 10%). The ASTM C642-13 standard was followed to have the sample’s density, absorption, and voids [89]. In addition, the ASTM D3171-99 standard was considered to obtain the fiber volume fractions of samples [90]. Equation (1) based on standard test ASTM C78/C78M-18 (using a simple beam with three or four-point loading) was considered to calculate the flexural strength (*σ_ps_*) [87]. Additionally, Equation (2) was employed to calculate initial stiffness (*I_s_*).
(1)$$\sigma_{ps}=P\times L/b\times d^{2}$$
*I_s_ = P_fc_/δ_fc_*(2)
where *b* is the sample width (mm), *d* is the sample thickness (mm), *L* is the support span (mm); *P* is the maximum applied load (N); *σ_ps_* is the flexural strength (MPa); *P_fc_* is the first crack load (N), *δ_fc_* is the first crack displacement (mm) and *I_s_* is the initial stiffness (N/mm).

A scanning electron microscope (SEM, LEO 440VR model, Oxford, UK) was used to analyze the morphology of the samples before and after the flexural test. The flexural samples’ macro-scale images were identified using a digital camera (Nikon D3000 10.2MP Digital SLR Camera with 18–55 mm f/3.5–5.6G AF-S DX VR Nikkor Zoom Lens, Tokyo, Japan). A steel ruler was employed to measure the crack lengths in the fractured flexural samples. A digital caliper (Newman, digital LCD, precision: 0.02 mm, resolution: 0.01 mm, CN) were utilized to measure the crack widths in the fractured flexural samples. A digital ruler was employed to measure the crack distance on the sample under four-point flexural load [91]. Off-axis pull-out test was applied after the flexural test for identifying the macro-level fiber-cement matrix bonding properties. This study will be published separately in the near future. To determine the energy absorption, the area under the entire load-deflection diagram was considered and calculated using MATLAB R2016a (The MathWorks, Inc., Natic, MA, USA). This was accomplished by applying the numerical integration and standard plotting tools of MATLAB [92].

## 3. Results and Discussion

### 3.1. Fiber Volume Fraction and Density Results 

Table 3 illustrates the panel density, fiber volume fractions (wt.%), void content, and water absorption results of the control (CC) and carbon fiber concretes (C-1D, C-2D, and C-4D). SEM micrographs of the multiaxis 3D concrete composites before the flexural test are exhibited in Figure 5a–i. As depicted in Table 3 and Figure 5, the total fiber content of all samples are between 2.48 and 2.56 wt.%. Additionally, all samples’ average density and void contents were 2.20 and 19.02%, respectively. The outside surrounding of the carbon fiber TOWs in the concrete structures was almost perfectly bonded in the C-1D (Figure 5a,b), C-2D (Figure 5d,e), and C-4D (Figure 5g,h) concretes. Thus, almost defect-free multiaxis 3D carbon concrete composite samples were made for the flexural test.

### 3.2. Flexural Test Results

After the flexural test data for the pristine (CC) and carbon concrete (C-1D-(0°), C-1D-(90°), C-1D-(+45°), C-2D-(0°), C-2D-(90°), C-2D-(+45°), C-4D-(0°), C-4D-(90°), C-4D-(+45°) and C-4D-(−45°)) obtained from the Shimadzu AG-X 100 (JP) testing machine, they were transferred to the Microsoft (MS) Excel spreadsheet 2013 via Trapezium^®^ software. Therefore, statistical computations on the raw data, including flexural first crack load and displacement, flexural maximum load and displacement, flexural first crack strength, and flexural maximum strength, as well as flexural stiffness, were carried out and analyzed as presented in Table 4. The standard deviations of these data were small due to defect-free manufacturing of various test samples. Figure 6a–c exhibits the load-displacement curves for the PAN carbon fiber concrete considering the neat concrete. Figure 7a–f shows stress-displacement curves for the C-1D and C-2D concrete structures. In addition, Figure 8a–d exhibits the stress-displacement curves for C-4D for principal and off-axis directions. Figure 9a,b illustrates the general characterization of the flexural load-displacement curves of the multiaxis 3D carbon fiber concrete composites (C-1D-(0°), C-1D-(90°), C-1D-(+45°), and C-2D-(−0°), C-2D-(−90°), C-2D-(+45°)) after the off-axis four-point flexural test. In addition, Figure 10 shows the general characterization of the flexural load-displacement curves of the samples (C-4D-(0°), C-4D-(90°), C-4D-(+45°) and C-4D-(−45°)) after the off-axis four-point flexural test.

In Figure 6a–c, the flexural load-displacement curves of the multiaxis 3D carbon concrete exhibited remarkably impressive results considering the pristine concrete, particularly due to the addition of high modulus carbon fibers in multiaxis (four) directions. It was identified that the flexural load displacement was heavily dependent on the fiber orientation. For instance, the flexural load of the C-1D-(0°) composite was higher than those of C-2D-(0°) and C-4D-(0°), and the C-2D-(0°) composite was also higher compared to C-4D-(0°) due to the high partial fiber weight fraction in the axial fiber (0° fiber) direction. Similar results were obtained for the 90° fiber, and ±45° fiber directions, and their flexural loads were proportion to their directional partial fiber weight fractions. When analyzing the load-displacement curves of the 3D multiaxis concrete composites, four distinct stages were identified as the elastic stage where the flexural load was proportional and rose linearly up to the first crack in the curve (Figure 6a–c). The multiple initial cracks stage, where multiple cracks in the cementitious matrix appeared, especially in ±bias concrete samples (Figure 6c). The strain hardening stage, where the fibers carried the load exponentially up to the maximum load via various mechanisms, such as fiber bridging, complex interphase load transferring through pull-out, and perhaps stick-slip, but it was less obvious in the 90° fiber direction (Figure 6b). The failure stage, where the catastrophic cementitious matrix and multiple filament breakages in the 3D fiber concrete occurred (Figure 6a–c).

In Figure 7a–f, the flexural stress-displacement curves of the C-1D-(0°) composite was higher than that of C-1D-(90°), C-1D-(+45°) and C-2D-(0°), C-2D-(90°) and C-2D-(+45°) due to the 0° fiber orientation and amount of partial fiber weight fraction in the axial fiber (0° fiber) direction. However, the flexural stress-displacement curves of C-1D-(90°), C-1D-(+45°), and C-2D-(+45°) showed similar curves compared to the control concrete.

In Figure 8a–d, the flexural stress-displacement curves of C-4D-(0°) exhibited similar behavior compared to C-4D-(±45°) except C-4D-(90°) in which it showed a slight strain hardening stage and more ductile material behavior. Material properties in the carbon fiber C-4D concrete were homogeneously distributed via multiaxis fiber orientation throughout the concrete structure.

As depicted in a general characterization of the load-displacement curves of C-1D, C-2D, and C-4D PAN carbon TOW concretes (Figure 9 and Figure 10), the fiber orientation influenced the flexural load-displacement curves of the sample considerably. When the 0° fiber changed its positions from the bottom (C-4D-(0°)) to the top of the concrete structure (C-4D-(90°)), the flexural load-displacement curves were shifted from strain hardening to the strain-softening (ductile) material stages. This was identified as a critical finding in the developed multiaxis 3D PAN carbon fiber concrete composites.

### 3.3. Flexural Load-Displacement

The average maximum flexural load, first crack load, maximum load displacement, and first crack displacement for the developed multiaxis 3D carbon concrete composites are shown in Figure 11. As illustrated in Figure 11 and Table 4, the maximum flexural load of the pristine concrete was 1374.93 N. The off-axis flexural maximum load of the carbon fiber concretes ranged from 920.16 to 3745.68 N for C-1D, from 1881.50 to 2644.95 N for C-2D, and from 1052.87 to 2799.47 N for C-4D. In addition, It was identified that standard deviations of the flexural maximum strength of all concrete structures were obtained between 0.43 and 1.84 (MPa), which was insignificantly small.

For the flexural maximum load in principal directions (0° and 90°), C-1D-(0°) was 41.62% higher than C-2D-(0°) and was 63.94% higher than for C-4D-(0°), and C2D-(0°) was 15.76% higher than C-4D-(0°). However, C-4D-(90°) was 88.82% lower than C-2D-(90°) and was 14.42% higher than C-1D-(90°), and C2-D-(90°) was 2.16 fold greater than C-1D-(90°). For the flexural maximum load in the bias directions (+45° and −45°), C-4D-(+45°) was 48.79% greater than C-2D-(+45°) and was 117.13% greater than C-1D-(+45°), and the C-2D-(+45°) was 45.93% higher than C-1D-(+45°). In addition, C-4D-(+45°) was 30.99% lower than C-4D-(−45°). It was found that the flexural load carrying performance on various developed carbon fiber concrete composites was proportional to the fiber orientation and amount of fiber in a particular direction. In addition, the flexural load-displacement of the C-1D-(0°), C-2D-(0° and 90° and +45°) and C-4D-(0° and ±45°) concrete composites were increased from 36.84 to 272.43% compared to the pristine concrete. The flexural performance of the concretes was influenced by the placement and orientation of the continuous PAN carbon fiber. In the first crack load (Figure 11), the first crack loads of all PAN carbon fiber concrete decreased by 5.78% compared to the neat concrete. However, the average first crack loads of the C-2D-(0° and +45°) PAN carbon fiber concrete composites were increased by 5.64% compared to the control concrete. Particularly, the average first crack load in the bias direction of the C-2D and C-4D composites was higher than the pristine concrete. The first crack loads of all the concrete structures changed proportionately with their corresponding maximum loads.

As depicted in Figure 11 and Table 4, the flexural maximum displacement of the neat (CC) concrete was 0.29 mm. The off-axis flexural maximum displacement of carbon fiber concretes ranged from 0.23 to 4.35 mm for C-1D, from 0.52 to 3.94 mm for C-2D, and from 0.31 to 4.77 mm for C-4D. The flexural displacement of C-4D was 57.14% greater than that of the C-2D and was 2.4 fold greater than C-1D. Moreover, the flexural displacement of the C-2D concrete was superior (52.73%) to the C-1D concrete. The flexural displacement of the C-4D, C-2D, and C-1D concrete was 13.66 fold, 8.69 fold, and 5.69 fold greater than the neat concrete. It was found that the placement and orientation of the continuous PAN-based carbon fiber in the pristine concrete noticeably influenced the displacement behavior of the concrete. Although the neat concrete showed a stiff and brittle behavior, adding carbon fibers to the neat concrete made it partly ductile and tough material. Yet, the first crack displacement of all continuous carbon concrete was increased by 31.04% compared to pristine concrete. It was identified that the first crack displacement of all carbon fiber concretes occurred proportionately with their flexural displacement.

### 3.4. Flexural Strength

Average flexural maximum strength, first crack strength, and initial stiffness for the developed multiaxis 3D carbon concrete composites are shown in Figure 12. As depicted in Figure 12 and Table 4, the average panel maximum strength of the control (CC) concrete was 6.19 MPa. The off-axis flexural maximum strength of the carbon fiber concretes ranged from 5.81 to 16.87 MPa for C-1D, from 8.47 to 11.91 MPa for C-2D, and from 4.74 to 12.61 MPa for C-4D concrete structure.

For the flexural maximum strength in the principal directions (0° and 90°), the C-1D-(0°) was 41.65% higher than C-2D-(0°) and was 63.95% higher than C-4D-(0°), and the C-2D-(0°) concrete composite was 15.74% higher than C-4D-(0°). However, the C-4D-(90°) concrete was 88.82% lower than C-2D-(90°) and was 14.49% higher than C-1D-(90°), and the C-2D-(90°) concrete was 2.16 fold greater than C-1D-(90°). For the flexural maximum strength in the bias direction (+45° and −45°), the C-4D-(+45°) was 44.28% greater than C-2D-(+45°) and was 2.17 fold greater than C-1D-(+45°), and C-2D-(+45°) was 45.78% higher than C-1D-(+45°). In addition, the C-4D-(+45°) composite was 31.08% greater than the C-4D-(−45°). It was found that the flexural strength performance of various developed carbon fiber concrete composites was proportional to the fiber orientation and amount of fiber in a particular direction, depending upon the structural fiber architecture. In addition, the flexural strength of the C-1D-(0°), C-2D-(0° and 90° and +45°) and C-4D-(0° and ±45°) concrete composites were 272.54 to 36.83% higher than that of the neat concrete, respectively. It was discovered that the placement and orientation of the continuous PAN-based carbon fiber in the neat concrete influenced the flexural strength performance of the concrete. As exhibited in Figure 12, the first crack strength of all PAN-based carbon fiber concrete composites was decreased by 5.82% compared to that of the neat concrete. However, the first crack strength of the C-2D-(0° and +45°) PAN carbon fiber concrete composites increased 5.65% compared to the control concrete. Particularly, the average first crack strength in bias direction, (+45°) of C-2D and C-4D were higher than the pristine concrete. It was found that the average first crack strength changed proportionately with the corresponding average maximum flexural strength of all continuous carbon fiber concretes. 

As shown in Figure 12 and Table 4, the average flexural initial stiffness of the control (CC) concrete was 27.33 MPa/mm. The off-axis flexural initial stiffness of the carbon fiber concretes ranged from 22.56 to 25.49 MPa/mm for C-1D, from 22.12 to 28.21 MPa/mm for C-2D, and from 20.41 to 23.84 MPa/mm for C-4D. The average flexural initial stiffness of the C-4D was 15.91% and 7.5% lower than the C-2D and C-1D concrete composite, respectively. Further, the flexural initial stiffness of the C-2D concrete was slightly higher (9.99%) than that of the C-1D concrete. The flexural initial stiffnesses of the C-4D and C-1D concretes were slightly (15.16 and 8.27%) less than the neat concrete, respectively. Additionally, the C-2D-(0 and +45°) was 9.1 and 8.1% greater than the neat concrete. It was contemplated that the initial stiffness properties of the concrete were probably affected by fiber orientations and structural architecture as well as fiber-matrix bonding and complex interlaminar load transfers. But, more research studies are required.

### 3.5. Angular (Off-Axis) Flexural Energy

Table 5 demonstrates average flexural energy for samples. The areas under the load-displacement curves of the multiaxis 3D carbon fiber concrete are typically illustrated in Figure 13 to define the flexural first crack energy, maximum load energy, strain hardening energy, strain-softening energy, and total energy. The angular (off-axis) flexural energy-displacement curves of multiaxis 3D carbon fiber concrete are exhibited in Figure 14a–c. Moreover, the angular flexural energy of the multiaxis 3D carbon fiber concretes is shown in Figure 15. As depicted in Figure 13, the energy was absorbed in deflecting a sample at a certain amount, and it is the area under a load-displacement curve in the flexural test. The area under the load-deflection curve of the multiaxis 3D carbon fiber concretes is defined by the following relations from Equations (3)–(7). It was calculated by using MATLAB R2016a (The MathWorks, Inc., Natick, MA, USA) using MATLAB’s numerical integration and standard plotting tools [92] where area calculation was mainly achieved by trapezoid method in numerical analysis technique via developed interface algorithm using the MATLAB codes.
(3)$$E_{ffc}=C_{0}C_{1}C_{2}$$
(4)$$E_{fml}=C_{0}C_{1}C_{3}C_{4}$$
(5)$$E_{fsh}=C_{2}C_{1}C_{3}C_{4}$$
(6)$$E_{fss}=C_{4}C_{3}C_{5}C_{6}$$
(7)$$E_{ft}=C_{0}C_{1}C_{3}C_{5}C_{6}$$
where, *E_ffc_* is the flexural first crack energy corresponding to the load-displacement in area *C*_0_*C*_1_*C*_2_; *E_fml_* is the flexural maximum load energy corresponding to the load-displacement in area *C*_0_*C*_1_*C*_3_*C*_4_; *E_fsh_* is the flexural strain hardening energy corresponding to the load-displacement in area *C*_2_*C*_1_*C*_3_*C*_4_; *E_fss_* is the flexural strain-softening energy corresponding to the load-displacement in area *C*_4_*C*_3_*C*_5_*C*_6_, and *E_ft_* is the flexural total energy corresponding to the load-displacement in area *C*_0_*C*_1_*C*_3_*C*_5_*C*_6_.

As illustrated in Figure 14a–c, the energy deflection curves steeply inclined in the order from neat concrete (CC) to multiaxis fiber (C-4D, C-2D, and C-1D) concretes, partly due to the strain hardening behavior of the carbon fibers in the concretes and partly carbon fiber properties. The C-1D-(0°), C-2D-(0°), and C-4D-(0° and 90° and ±45°) structures demonstrated outstandingly better energy absorption performance with regards to the C-1D-(90° and +45°), C-2D-(+45°) and CC structures probably due to the strain hardening energy absorption stage and fiber orientation. It was observed that C-1D-(90° and +45°), C-2D-(+45°), and C-4D-(90°) had an insignificant strain hardening stage (Figure 6b,c). The off-axis fiber energy absorption in C-4D was significant considering the C-2D and C-1D structures due to the multiaxis fiber placement and homogeneous fiber distribution throughout the in-plane of the concrete structure.

As exhibited in Figure 15 and Table 5, the flexural maximum load energy of the neat (CC) concrete was 0.16 J. The flexural maximum load energy of the PAN-based carbon fiber concretes (C-1D, C-2D, and C-4D) ranged from 0.09 to 11.23 J. The flexural maximum load energy in the principal directions (0° and 90°), the C-1D-(0°) was 81.42% higher than C-2D-(0°) and was 2.8 fold higher than C-4D-(0°), and C2D-(0°) was 54.75% higher than C-4D-(0°). However, the flexural maximum load energy of C-2D-(90°) concrete composite was extraordinarily higher compared to the C-4D-(90°) and C-1D-(90°) concretes, and C-4D-(90°) was 22.22% higher than C-1D-(90°). For the flexural maximum load energy in the bias directions (+45 and −45°), the C-4D(+45°) concrete was 26.96 fold higher than C-2D-(+45°) and was 43.81 fold higher than C-1D-(+45°), and C-2D-(+45°) was 62.50% higher than C-1D-(+45°). In addition, the C-4D-(+45°) was 3.85% higher than C-4D-(−45°). It was found that the flexural energy absorption performance of the various developed carbon fiber concrete composites was proportional to the fiber orientation and amount of fiber, in particular, the fiber direction depending upon structural fiber architecture. Additionally, the C-1D-(0°), C-2D-(0 and 90°), and C-4D-(0 and ±45°) concretes were outstandingly superior to the control concrete. It was identified that the flexural energy absorption performance of the fiber concrete was influenced by the parameters of fiber placement and fiber orientation. In the first crack flexural energy, as depicted in Figure 15, the first crack energy of C-1D was equal to the pristine concrete. However, the average first crack energy of the C-2D and C-4D concrete composites increased 6.25% compared to that of the control concrete. Particularly, the average first crack energies in the bias direction of C-2D and C-4D were greater than that of the pristine concrete. 

For the PAN-based carbon fiber concretes (C-1D, C-2D, and C-4D), the flexural strain hardening energy ranged from 3.88 to 11.01 J. Considering the flexural strain hardening energy in the principal directions (0 and 90°), the C-1D-(0°) concrete was 82.59% higher than C-2D-(0°) and was 2.84 fold higher than C-4D-(0°), and the C-2D-(0°) composite was 55.41% higher than C-4D-(0°). However, the C-2D-(90°) concrete was extraordinarily higher compared to C-4D-(90°), and the C-1D-(90°) had no strain hardening energy stage. For the flexural strain hardening energy in the bias directions (+45° and −45°), the C-4D-(+45°) was 2.28% higher than C-4D-(−45°). However, the C-2D-(+45°) and C-1D-(+45°) composites had no strain hardening energy stages. It was found that the flexural strain hardening energy absorption performance on various developed carbon fiber concrete composites was proportional to the fiber orientation and amount of fiber in a particular direction, depending upon the structural fiber architecture. In addition, the C-1D-(0°), C-2D-(0° and 90°), and C-4D-(0° and ±45°) concrete composites were extraordinarily higher compared to the pristine concrete. 

### 3.6. Off-Axis FTIR Analysis after Flexural Test

The Fourier transform infra-red (FTIR) spectra of the CC (pristine concrete), PAN-based carbon fiber, C-1D, C-2D, and C-4D carbon fiber reinforced concretes are shown in Figure 16. In addition, the FTIR spectra of the CC (pristine concrete), PAN-based carbon fiber, and four C-4D carbon fiber concrete composites such as C-4D warp cross-section, C-4D filling cross-section, C-4D +bias cross-section, and C-4D-bias cross-sectional samples are exhibited in Figure 17.

In the FTIR spectrum of the pristine concrete (CC), as demonstrated in Figure 16, the main compound detected in the CC was calcium carbonate, supported by several strong bands. Meanwhile, 1423 cm^−1^ was assigned to the C–O stretching vibration. Concerning the other parts of the spectrum, the band at 1083 cm^−1^ may be attributed to Si-O asymmetric stretching vibrations [93]. The bands at 868 and 771 cm^−1^ can be attributed to the -C-C-, -C-H- groups [94,95,96]. The peak extending between 3500 and 3000 cm^−1^ was the absorption peak of the Si-OH and –OH groups. In the FTIR spectrum of the PAN carbon fiber (C), as illustrated in Figure 16, the peaks at 3022 and 2952 cm^−1^ were the deformation vibrations of the -CH2-CH- (aliphatic) groups, whereas the peak at 2336 cm^−1^ was ascribed to the deformation vibrations of the functional group of −C≡N (nitryl) [97]. The peaks at 1432 cm^−1^ and 1356 cm^−1^ were attributed to the stretching vibrations of the -C=N- group. In addition, the peaks observed between 1194 and 885 cm^−1^ corresponded to -C-C- and -C-N- bonds [98]. In the FTIR spectrum of the C1-D (warp cross-section), C-2D (filling cross-section), and C-4D (warp cross-section) PAN carbon/cementitious concrete composites (Figure 16), a common signal on all developed structures were detected. The peak observed between 3500 and 3000 cm^−1^ bands may be attributed to the deformation vibrations of H_2_O, polycarboxylate, -Si-OH, and -Ca-OH groups. In the 3000–2850 cm^−1^ band spacing, the deformation vibrations of -CH2-CH- (aliphatic) groups were observed, whereas the signal between 2300 and 2200 cm^−1^ corresponded to the −C≡N (nitryl) group. The signals between 1600 and 1300 cm^−1^ were attributed to stretching vibrations of the -C=C-, -C=O, and -C=N groups. The signal observed between 1100 and 1000 cm^−1^ was assigned to Si-O asymmetric stretching vibrations, and another signal in the 900–500 cm^−1^ band spacing was considered the stretching vibrations of single bond groups.

In the FTIR spectrum of C-4D (-bias cross-section), C-4D (+bias cross-section), C-4D (filling cross-section), and C-4D (warp cross-section) as exhibited in Figure 17, almost similar results were obtained compared to the C-1D (warp cross-section) and C-2D (filling cross-section) (Figure 16). It was considered that the fiber placement and orientation in the concrete did not affect their FTIR spectrum. From the FTIR analysis, it was identified comparatively that probably a strong physical interaction between the cementitious components and PAN carbon fiber (filament TOWs) was realized. These findings were supported by fractured sample SEM images (Figures 22 and 23). Hence, their directional flexural properties were homogeneously distributed throughout the concrete structure. As a result, the C-4D structure controlled the number of cracks, crack width, crack distance, and it improved the flexural energy absorption capacity of the concrete structure. However, more study is required, especially regarding the cement-fiber interlaminar regions in the multiaxis 3D carbon fiber concrete composites.

### 3.7. Flexural Failure Results

Table 6 outlines the flexural failure results. The length and number of cracks in the fractured samples are exhibited in Figure 18. Further, the width and number of cracks in fractured multiaxis 3D fiber concrete are shown in Figure 19. As shown in Table 6 and Figure 18 and Figure 19, the pristine concrete exhibited catastrophic failure under the four-point flexural load. The off-axis flexural crack width of the carbon fiber concretes ranged from 0.217 to 0.585 mm for C-1D, from 0.216 to 0.630 mm for C-2D, and from 0.461 to 2.102 mm for C-4D. The crack distances in C-1D-(0°), C-2D-(0°), and C-2D-(90°) were 103.80, 88.60, and 68.90 mm, respectively. However, the crack distance ranged from 52.40 to 68.10 mm for C-4D-(0°), from 37.30 to 73.90 mm for C-4D-(+45°), and from 53.30 to 78.10 mm for C-4D-(−45°).

Various fractured multiaxis 3D carbon fiber concretes including C-1D-(0°, 90° and +45°), C-2D-(0°, 90°, and +45°), and C-4D-(0°, 90°, +45°, and −45°), as well as the neat concretes on the top, bottom, and cross-sections and their failure modes after the off-axis flexural test are exhibited in Figure 20. For the flexural crack width in principle directions (0° and 90°), the C-1D-(0°) concrete was 7.14 and 4.41% less than that of the C-2D-(0°) and the C-4D-(0°) composites, respectively. Additionally, C-2D-(0°) was 2.94% greater than C-4D-(0°). On the contrary, the crack width in C-2D-(90°) was smaller than that of C-4D-(90°), which was near to catastrophic failure, and C-1D-(90°), which exhibited complete catastrophic failure. For flexural crack width in the bias direction (+45° and −45°), the C-4D-(+45°) concrete was 113.43 and 112.44% greater than the C-2D-(+45°) and the C-1D-(+45°) composites, respectively. C-2D-(+45°) crack width was almost equal to C-1D-(+45°). However, C-4D-(+45°) was 48.59% less than C-4D-(−45°). The in-plane crack length in all the developed 3D carbon fiber concrete composites was equal (75 mm). Crack distance in C-4-D-(0, +45, and −45°) was smaller than C-2D-(0 and 90°) and C-1D-(0°) due to the number of cracks. Yet, the numbers of cracks in C-4D and to some degree in C-2D were greater and multiaxially distributed compared to C-1D. We summarized that multiaxis 3D carbon fiber concretes, especially the C-4D structure, controlled the crack number, crack width, and crack distance during four-point flexural loading. However, more research is needed to identify the in-plane and through-the-thickness crack initiation and propagation under off-axis loading.

### 3.8. Flexural Failure Analysis

The top, bottom, front and back views of the fractured concrete and their failure modes after the off-axis flexural test are illustrated in Figure 20. Figure 21a–i shows some of the pull-out of samples in C-1D-(0 and +45°), C-2D-(0 and 90°), and C-4D-(90 and +45°) after the off-axis flexural test.

As shown in Figure 20, on the top and bottom surfaces of the neat concrete (CC) structure, in-plane fatal crack failure was received, and it demonstrated a fragile attitude. The neat concrete entirely failed under the flexural loading. The CC structure exhibited total through-the-thickness cracking on the front and backside. The in-plane top (compression side) and bottom (tension side) surfaces of the C-1D-(0 and +45°) structure had minor flexural cracks normal to the warp carbon TOW (0°), and minor in-plane shears were identified. In the out-of-plane back and front sides, flexural failure was initiated. Later, it was deflected to shear failure, probably owing to the different modulus of the PAN fiber and matrix. After debonding close to the outer surface of the filament bundle, crack initiation was possibly started, and it propagated until the through-the-thickness failure was completed to where multiple warps (0 and +45°) carbon TOWs were not severely damaged. Only minor filament breakages were identified during multiple off-axis pull-outs after the four-point flexural test, as shown in Figure 21a–c. The developed off-axis pull-out test after the flexural test for this purpose is planned for a separate paper to be published in the near future. The in-plane top (compression side), bottom (tension side) surfaces, and the out-of-plane through-the-thickness of the C-1D-(90°) structure showed catastrophic flexural crack failures due to a parallel loading direction in the warp fiber. The in-plane top (compression side), bottom (tension side) surfaces, and the out-of-plane through-the-thickness of the C-2D-(0, 90 and +45°) structure had minor flexural cracks where multiple warps and filling (0 and 90°) carbon TOWs were not harshly damaged. Only minor filament breakages were observed during multiple off-axis pull-outs after the flexural test, as shown in Figure 21d–f. The in-plane top (compression side), bottom (tension side) surfaces, and the out-of-plane through-the-thickness of the C-4D-(0 and ±45°) structure had minor flexural cracks where multiple warps and filling (0, 90, and ±45°) carbon TOWs were not severely damaged. Only minor filament breakages were observed during the multiple off-axis pull-outs after the flexural test, as shown in Figure 21g–i. The in-plane top (compression side), bottom (tension side) surfaces, and the out-of-plane through-the-thickness of the C-4D-(90°) structure showed large flexural cracks opening due to fiber placement and orientation in the structure architecture.

Probably, the fracture mechanism comprised regional matrix fracture, matrix-filament bundle debonding for each yarn set in the concrete, filament TOW bridging due to tensile strengthening via principal fibers (0°), partly complex shearing via particularly angular yarn orientations, and shortly after outer partial or complete filament TOWs pull-out and stick-slip based on telescopic movement principles between the outer filament-matrix or core filament-filament frictions. As a result, all yarns carried the flexural load through complex load distribution of the cementitious matrix, such as the warp, filling, and ± bias yarns. Nevertheless, more research is required to define the complex crack control contribution, including some types of stress concentration in the through-the-thickness of the fiber concrete considering each fiber set during flexural loading.

### 3.9. Failure Analysis by Scanning Electron Microscope (SEM) Micrograph

Figure 22a–d exhibits the SEM graph of the inside fractured surface of the C-1D and C-2D concretes. The cementitious matrix did not penetrate the core carbon filament bundles in the C-1D concrete entirely because of the high filament packing density (Figure 22a). Around the fiber-matrix boundary zone, fractured matrix particles in the filament bundle surface and multiple filamentary splitting were observed (Figure 22a,b). Multiple inter and intra-filament bundle-matrix delamination’s near the boundary region at the carbon warp (0°) and multiple brittle matrix breakages were found (Figure 22c). At the boundary of the cementitious matrix-filament of the C-2D concrete (Figure 22d), slight shear hackle mark on the matrix, large filament bundle/matrix splitting, and multiple failures in the filament-matrix at the filling (90°) were identified. 

Figure 23a–d illustrates the SEM graph of the inside fractured surfaces of the C-4D concrete. Filament pull-out traces and multiple macro matrix breakages on the carbon warp (0°) of the C-4D concrete were found (Figure 23a). Broom like-damage features and brittle matrix failures were realized. Filament pull-out traces on the carbon filling (90°) of the C-4D concrete and filament-matrix debonding were pointed out (Figure 23b). Minor failed matrix particles on the filament surfaces in the bias (+45°) carbon TOW and minor splitting filaments were detected. In contrast, multiple brittle matrix-filament breakages near the outer region of bias (−45°) fiber TOWs were identified (Figure 23c). Additionally, near to the core of the C-4D-(−45°), minor filament entanglement and multiple splitting filaments were found where the core of the filament bundle was not penetrated by the cementitious matrix (Figure 23d). 

## 4. Implementation of Research Outcomes in Real-Life Projects

Multiaxis 3D carbon/cementitious fiber concretes which were developed for the study, can find several applications such as prefabricated panels, bridge decks, industrial floors, and airport runways. In addition, they can be used for rehabilitation purposes, such as repairing or retrofitting in aging infrastructure applications. Some of the examples of such applications are seismic retrofit, confinement, connectors for numerous settlements, including industrial and nuclear plants and parking structures, bridge decks, and seawall construction [99]. This is probably due to their better damage-tolerant material properties based on critical multiaxially load-carrying capacity. This could also be attributed to their structural architecture and fiber TOW’s continuous form and control of the multiple microcrack formation during complex loading. Moreover, fiber-reinforced polymer (FRP) concrete owing to its nonconductive properties, became the promising candidate for the facilities with magnetic resonance imaging (MRI) equipment and the electronics laboratories [99].

## 5. Conclusions

Multiaxis three-dimensional polymeric PAN carbon fiber/cementitious matrix concretes were developed. The flexural strength of the C-1D-(0°), C-2D-(0°, 90°, and +45°), and C-4D-(0° and ±45°) concrete composites were considerably higher compared to the pristine concrete. The flexural strength performances for various developed polymeric carbon fiber concrete composites were generally proportional to the fiber orientation and amount of fiber in a particular direction, depending upon the structural fiber architecture. On the other hand, the flexural energy absorption performance of the C-1D-(0°), C-2D-(0° and 90°), and C-4D-(0 and ±45°) concrete composites were outstandingly superior to the neat concrete.

Neat concrete exhibited fatal brittle crack failure behavior. On the contrary, the fracture mechanisms in the four directional carbon fiber/cementitious matrix concretes were described as local brittle matrix breakages and broom like-damage features on the filaments, filament bundle-matrix debonding for each yarn set, multifilament bridging in the TOW (perhaps particle bridging in the matrix) due to tensile strengthening and rapid pulling out of the outer filaments and stick-slip between outer filament-matrix or core filament-filament friction. This likely resulted in many micro-cracks in the C-4D concretes, which controlled the number of cracks, crack width, and crack distance. Therefore, the multiaxis 3D fiber polymeric concrete was contemplated damage-tolerant materials compared to the neat concrete. 

More research is required in future endeavors to model the in-plane and out-of-plane crack initiation and propagation under angular (off-axis) flexural loading for precise designing such structures. 

## Figures and Tables

**Figure 1 polymers-13-03073-f001:**
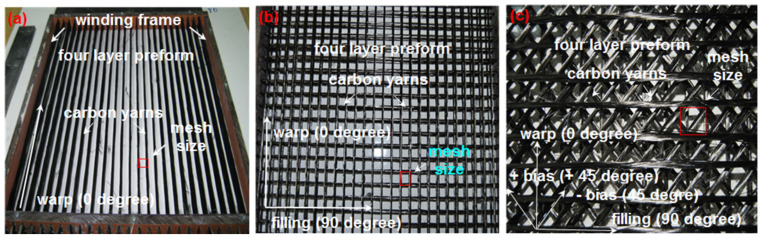
Various multiaxis 3D carbon preforms for concrete structures produced by “winding and molding” method. (**a**) Unidirectional preform (C-1D); (**b**) biaxial preform (C-2D); (**c**) multiaxis preform (C-4D) (digital image).

**Figure 2 polymers-13-03073-f002:**
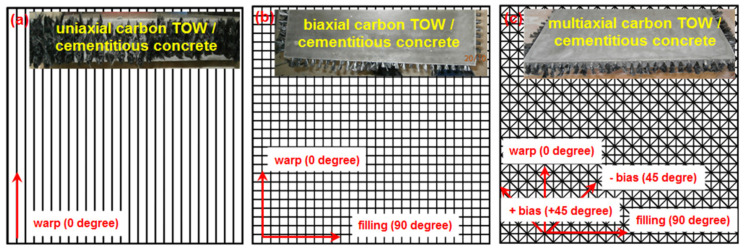
Schematic and actual view of multiaxis 3D carbon fiber concrete composites. (**a**) uniaxial (0°) (C-1D); (**b**) biaxial (0°/90°) (C-2D) and (**c**) four direction (90°/±45°/0°) (C-4D) (digital image).

**Figure 3 polymers-13-03073-f003:**
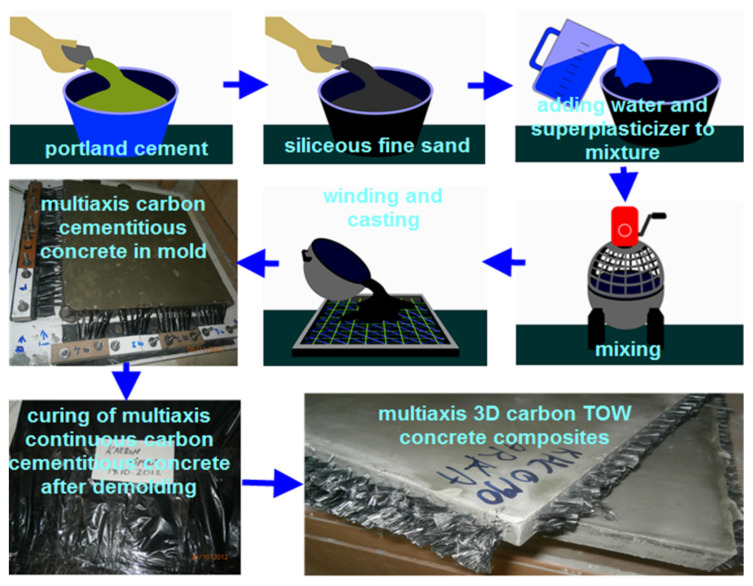
Processing route of multiaxis 3D carbon/cementitious concretes, schematic and actual.

**Figure 4 polymers-13-03073-f004:**
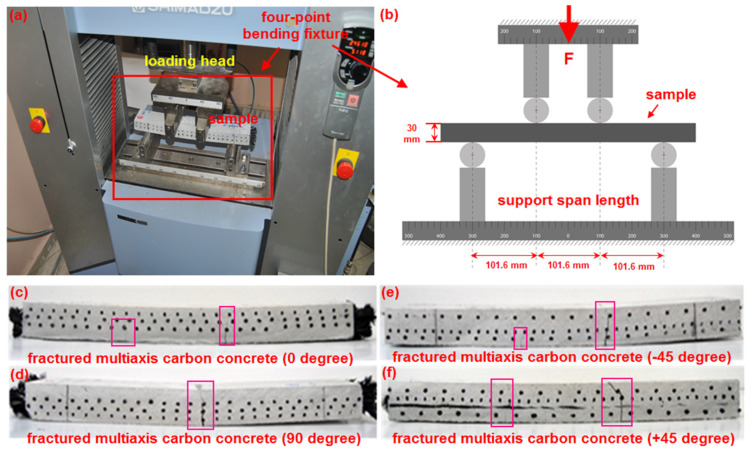
Multiaxis 3D carbon concretes during the flexural test. (**a**) Pictorial view of the flexural test of sample in the universal testing machine; (**b**) specimens with a four-point flexural test fixture, schematic; (**c**) fractured carbon concrete, C-4D-(0°); (**d**) C-4D-(90°); (**e**) C-4D-(+45°), and (**f**) C-4D-(−45°) (digital image).

**Figure 5 polymers-13-03073-f005:**
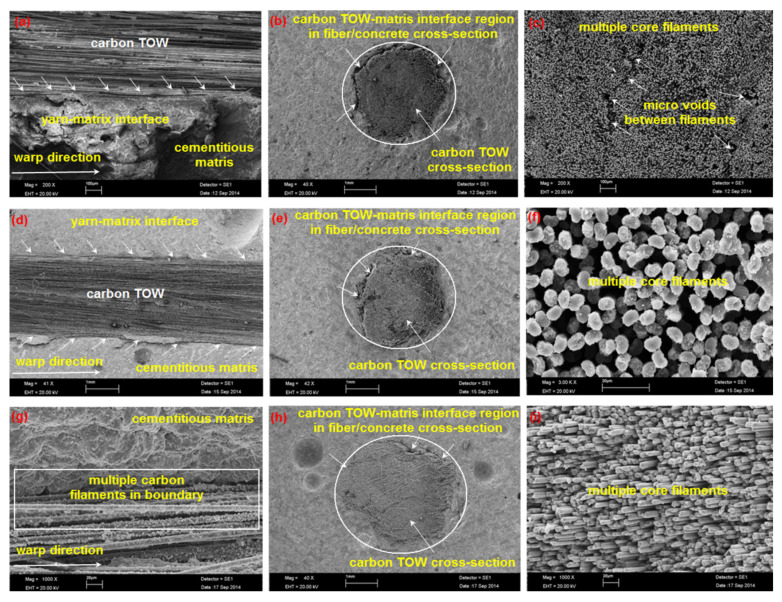
SEM graph of some multiaxis 3D carbon fiber concrete before the flexural test. (**a**) C-1D structure in carbon fiber length; (**b**) C-1D structure in filament bundle cross-section; (**c**) carbon filament bundle core of C-1D; (**d**) C-2D structure in filling carbon fiber length; (**e**) C-2D structure in filling carbon TOW cross-section; (**f**) close SEM images of the filament bundle core of C-2D; (**g**) C-4D structure in filling carbon fiber length; (**h**) C-4D structure in filling filament bundle cross-section; (**i**) close views of filament TOW core of C-4D.

**Figure 6 polymers-13-03073-f006:**
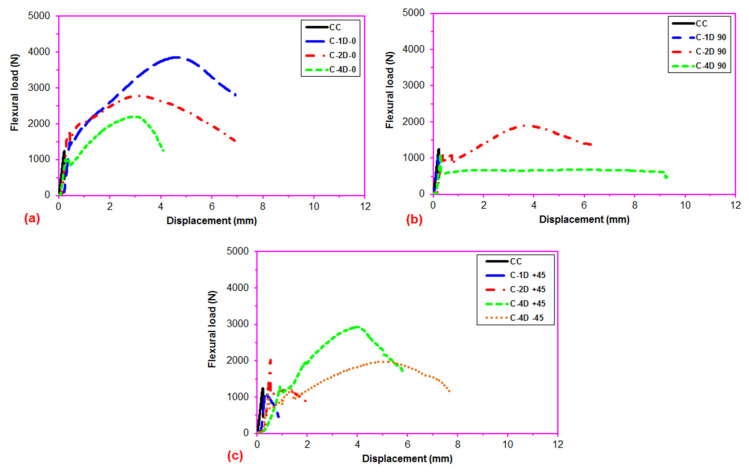
Flexural load-displacement curves of the multiaxis 3D carbon fiber concrete. (**a**) Various carbon TOW concrete structures at 0 degrees flexural test; (**b**) various carbon TOW concrete structures at 90 degrees flexural test; and (**c**) various carbon TOW concrete structures at bias directions flexural test.

**Figure 7 polymers-13-03073-f007:**
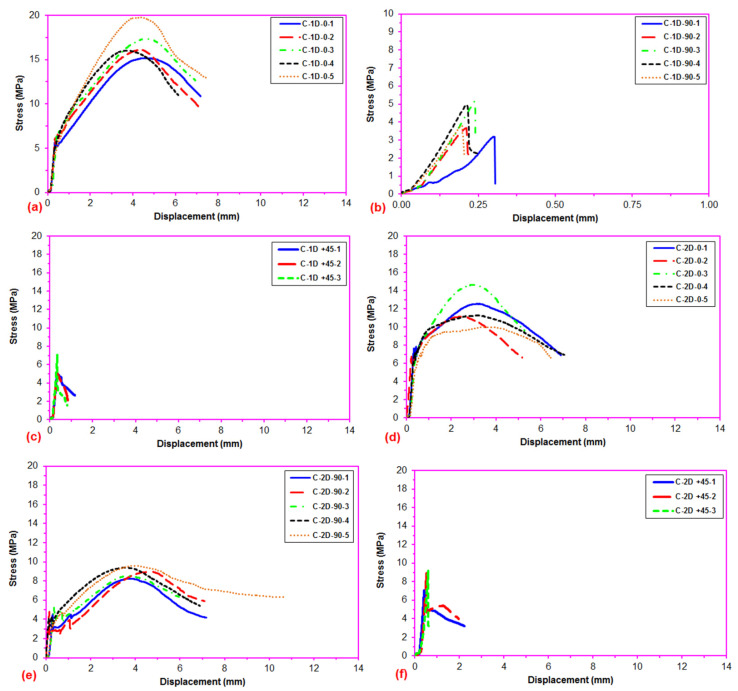
Stress-displacement curves of uniaxial and biaxial 3D concrete composites in the four-point flexural test. (**a**) C-1D at 0 degree flexural test; (**b**) C-1D at 90 degree flexural test; (**c**) C-1D at +45 direction flexural test; (**d**) C-2D at 0 degree flexural test; (**e**) C-2D at 90 degree flexural test; and (**f**) C-2D at +45 direction flexural test.

**Figure 8 polymers-13-03073-f008:**
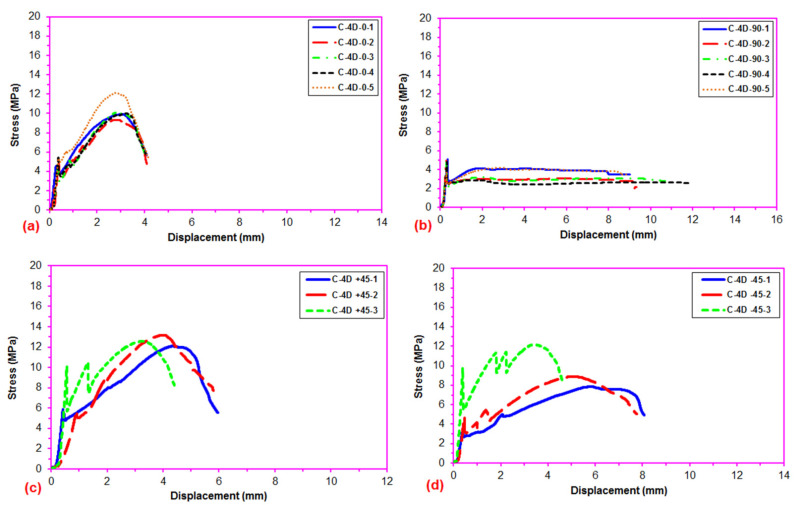
Stress-displacement curves of multiaxis 3D carbon fiber concretes. (**a**) C-4D at 0 degree flexural test; (**b**) C-4D at 90 degree flexural test; (**c**) C-4D at +45 direction flexural test; (**d**) C-4D at −45 direction flexural test.

**Figure 9 polymers-13-03073-f009:**
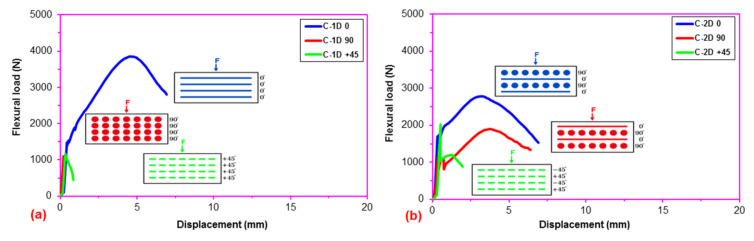
General characterization of off-axis flexural load-displacement curves of the multiaxis 3D carbon fiber concretes. (**a**) C-1D concrete and (**b**) C-2D concrete composite.

**Figure 10 polymers-13-03073-f010:**
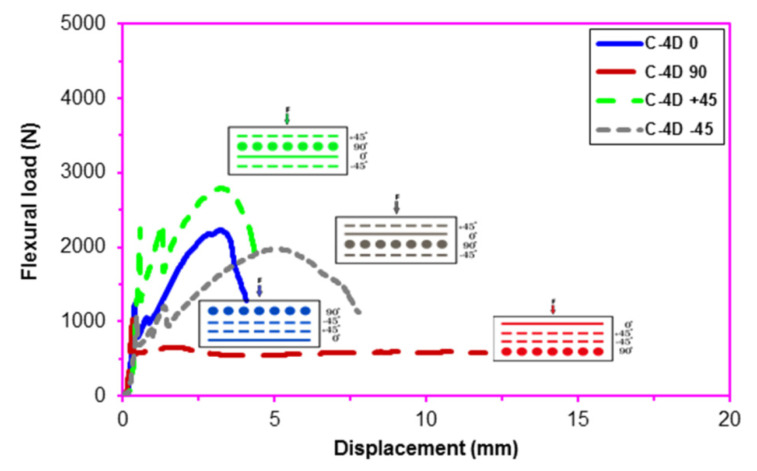
General characterization of the off-axis flexural load-displacement curves of C-4D concrete composite.

**Figure 11 polymers-13-03073-f011:**
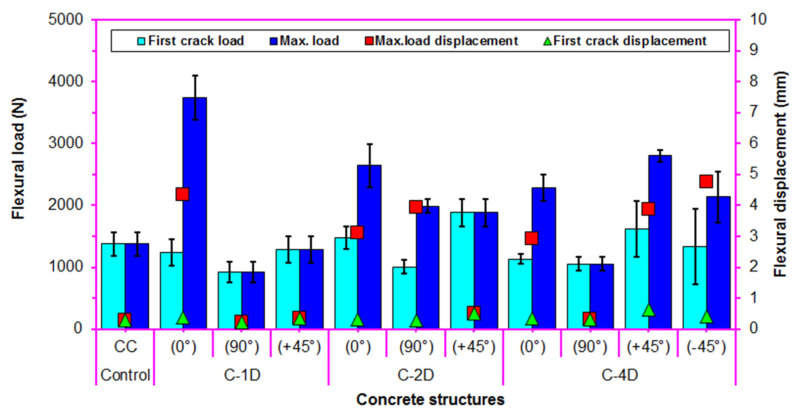
Average maximum and first crack load-displacements for multiaxis 3D carbon fiber concrete.

**Figure 12 polymers-13-03073-f012:**
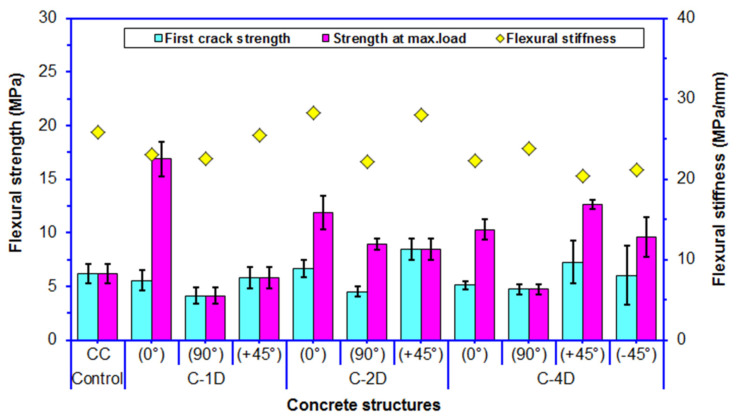
Off-axis flexural strength, first crack strength, and initial stiffness of various multiaxis 3D fiber concrete.

**Figure 13 polymers-13-03073-f013:**
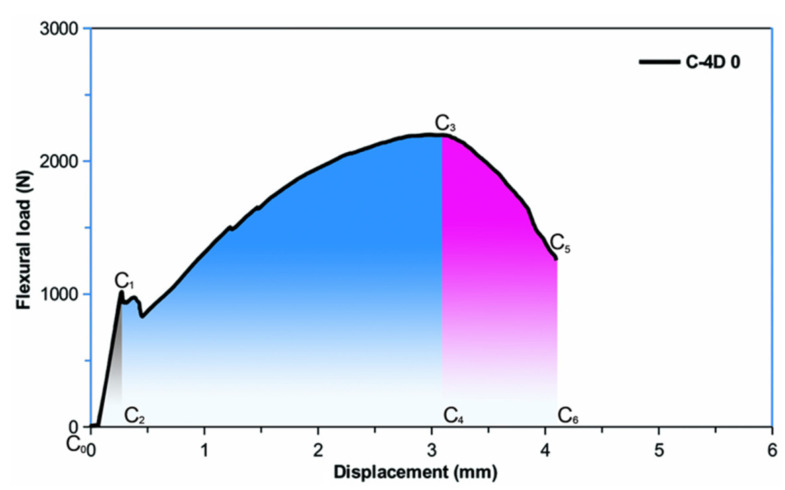
Presentation of the representative areas under the load-deflection curves of the multiaxis 3D carbon fiber concrete to calculate various flexural energies.

**Figure 14 polymers-13-03073-f014:**
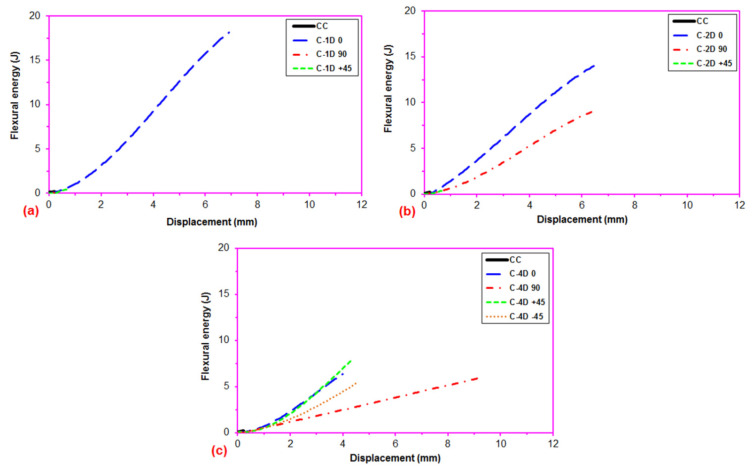
Flexural energy-displacement curves of the multiaxis 3D concrete after the off-axis four-point flexural test. (**a**) C-1D concrete composites; (**b**) C-2D concrete composites and (**c**) C-4D concrete composites.

**Figure 15 polymers-13-03073-f015:**
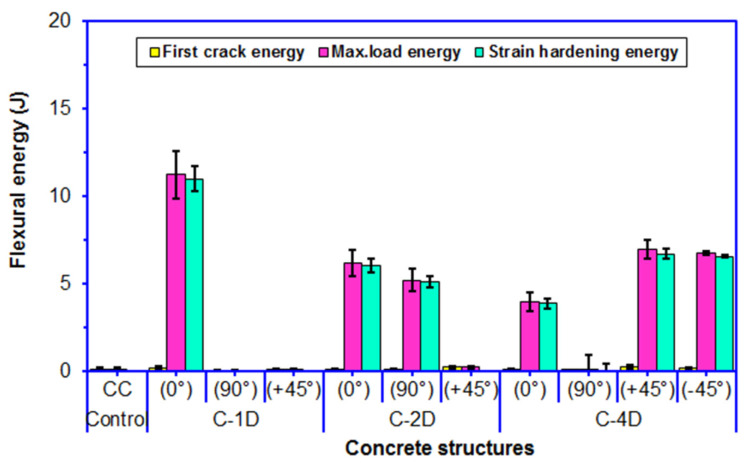
Off-axis flexural energy, first crack energy, and strain hardening energy of multiaxis 3D carbon fiber concrete.

**Figure 16 polymers-13-03073-f016:**
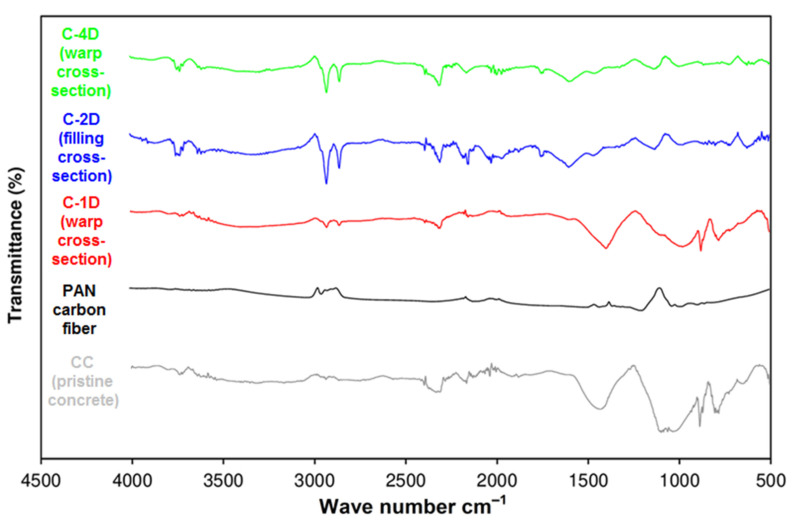
Fourier transform infra-red spectra of CC (pristine concrete), PAN carbon fiber, C-1D, C-2D, and C-4D carbon fiber concrete composites.

**Figure 17 polymers-13-03073-f017:**
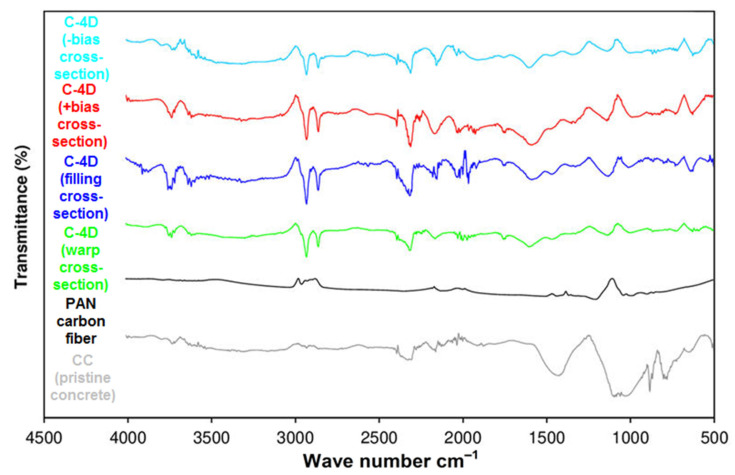
Fourier transform infra-red spectra of the CC (pristine concrete), PAN carbon fiber, and C-4D carbon fiber concrete composites.

**Figure 18 polymers-13-03073-f018:**
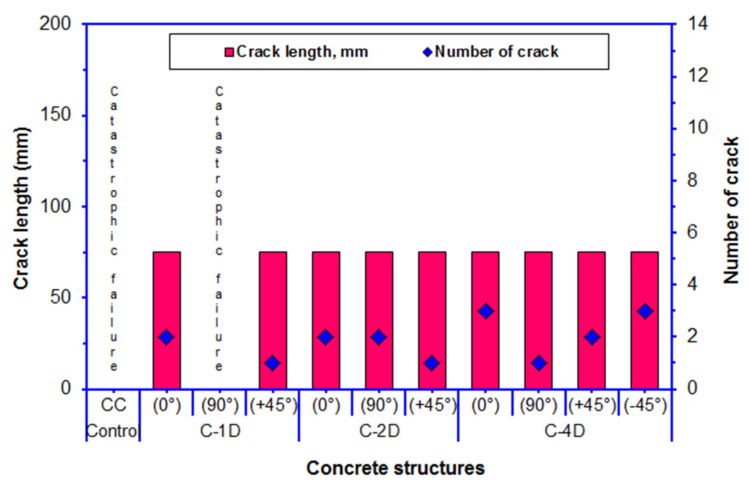
The number of cracks and their length in the failed samples after the off-axis four-point flexural test.

**Figure 19 polymers-13-03073-f019:**
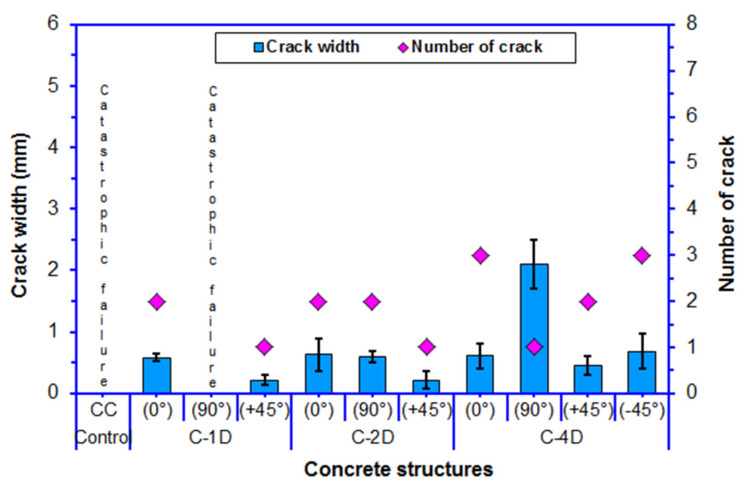
The number of Cracks and their width in the failed samples after the off-axis four-point flexural test.

**Figure 20 polymers-13-03073-f020:**
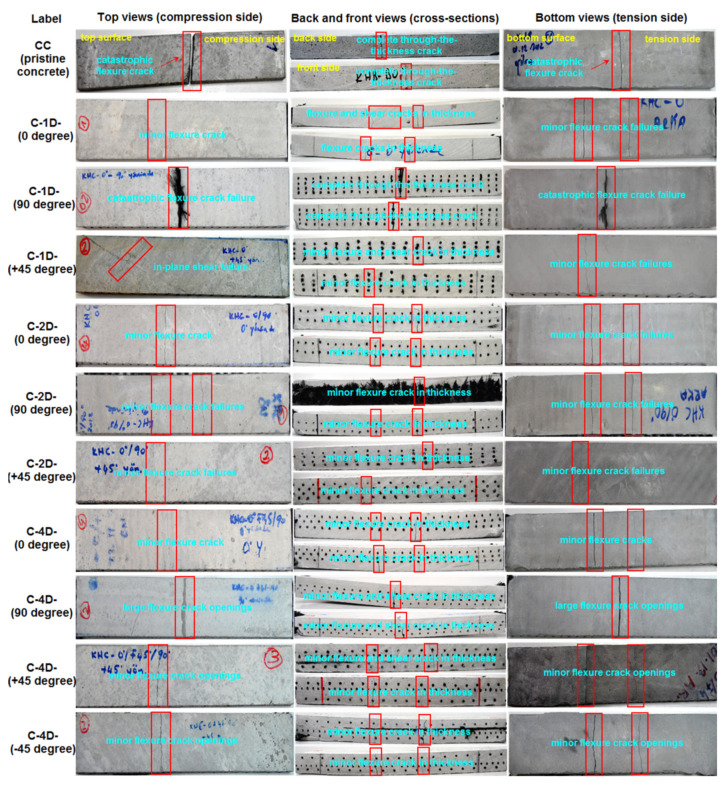
Various fractured multiaxis 3D carbon fibers, including the neat concretes top, bottom, cross-sections, and failure modes after the off-axis flexural test (Digital photo image).

**Figure 21 polymers-13-03073-f021:**
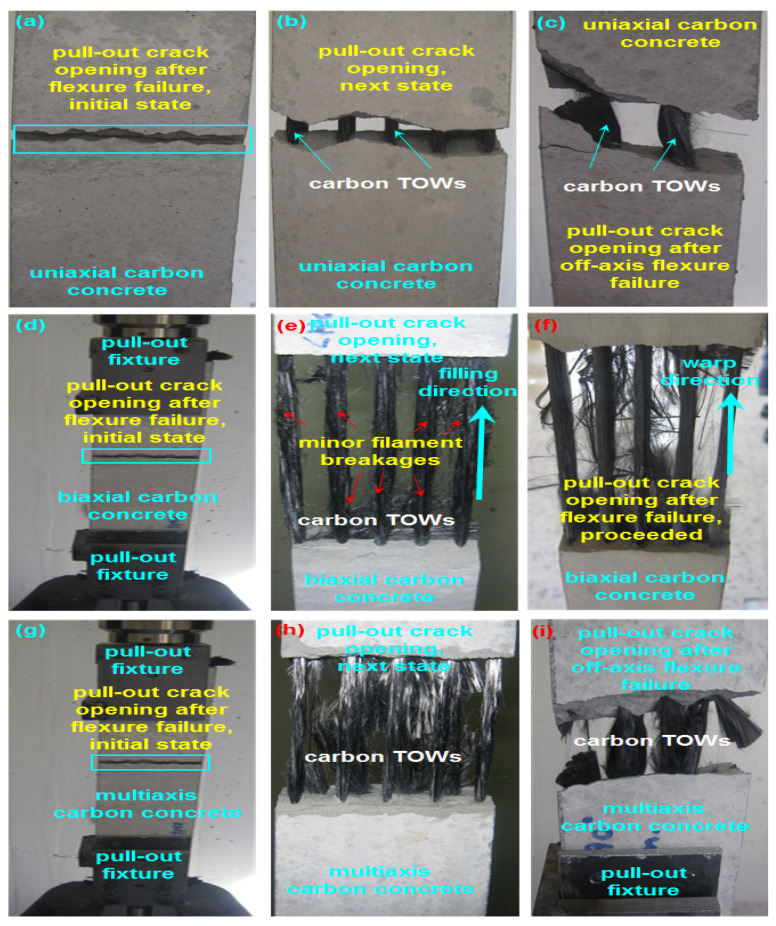
Pull-out of multiaxis 3D carbon fiber concretes. (**a**) Initial state of pull-out crack opening after flexural failure (C-1D); (**b**) next state of pull-out crack opening (C-1D); (**c**) pull-out crack opening after off-axis flexural failure (C-1D); (**d**) initial state of pull-out crack opening after flexural failure in filling (C-2D); (**e**) next step of pull-out crack opening in filling (C-2D); (**f**) warp direction pull-out crack opening after flexural failure, proceeded (C-2D); (**g**) initial state of pull-out crack opening after flexural failure in filling (C-4D); (**h**) next state of pull-out crack opening in filling (C-4D) and (**i**) pull-out crack opening after off-axis flexural failure in the warp (C-4D).

**Figure 22 polymers-13-03073-f022:**
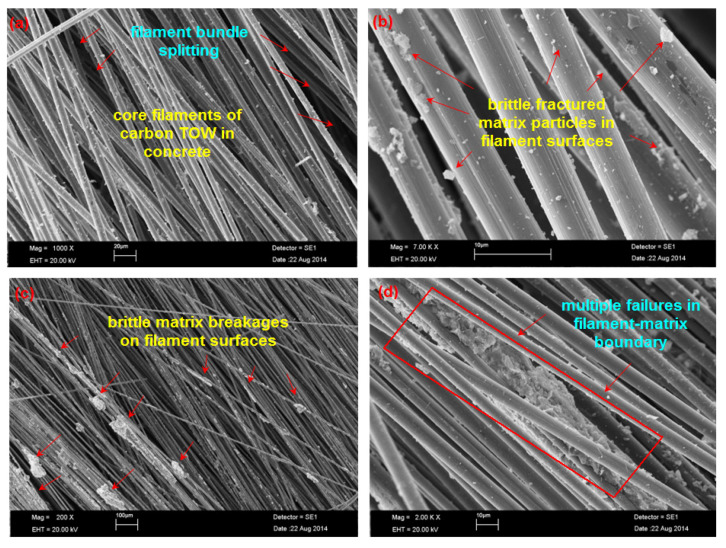
SEM graph of the fractured samples. (**a**) Failed fiber TOW in C-1D; (**b**) enlarge view of the failed fiber-cementitious matrix in C-1D; (**c**) failed matrix in the warp (0°) near the boundary region in C-2D; (**d**) failed filling (90°) near the boundary region in the C-2D concrete.

**Figure 23 polymers-13-03073-f023:**
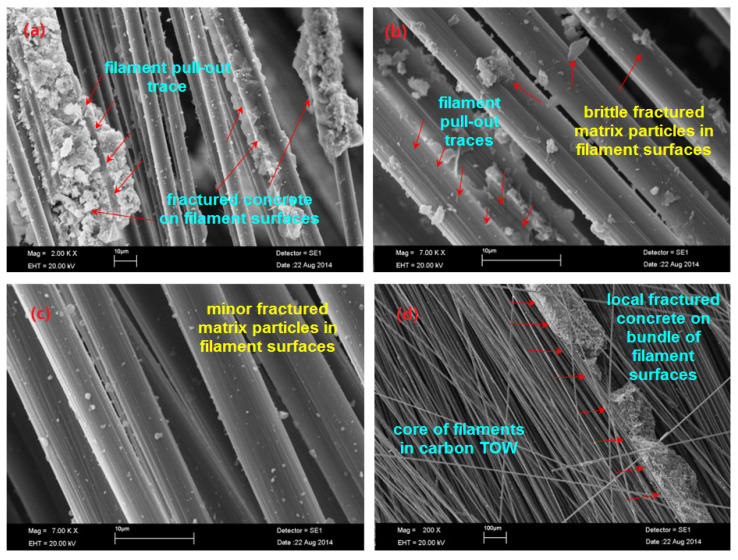
SEM graph of fractured samples. (**a**) Failed surface of the carbon warp (0°) in the C-4D; (**b**) enlarged view of the failed fiber-cementitious matrix in carbon filling (90°) in C-4D; (**c**) fractured matrix in bias fiber (+45°) in C-4D; (**d**) fractured matrix in the bias fiber (−45°) in the C-4D concrete.

**Table 1 polymers-13-03073-t001:** Specifications of continuous carbon fiber utilized in the concrete composites [81].

Fiber	Fiber Diameter (μm)	Linear Density (tex)	Density (g/cm^3^)	Dry Fiber Tenacity (cN/tex)	Tensile Strength (MPa)	Tensile Modulus (GPa)	Breaking Elongation (%)	Melting Temperature (C°)
PAN Carbon (AKSACA, DowAksa, Yalova, Turkey)	6	1600	1.76	≥77 (≥1355.2 MPa)	4200	240	1.80	>1200

**Table 2 polymers-13-03073-t002:** Specifications of multiaxis 3D concrete structures for the concrete composites.

Type of Structure	Label	Fiber Type	Yarn Ends	Weave Type	Stacking Sequence /Number of Layers	Number of Twisting (Turn/m)	Plied Yarn (tex)	Density of Preform (End/15 cm)	Mesh Opening (mm)
Control	CC	Concrete	-	-	-	-	-	-	-
Carbon concrete	C-1D-(0°)	Carbon	4	Non-interlaced	[0°]_4_	10	6400	10	15
	C-1D-(90°)				[90°]_4_				
	C-1D-(+45°)				[+45°]_4_				
	C-2D-(0°)	Carbon	4	Non-interlaced	[90°/0°]_2_	10	6400	10	15 × 15
	C-2D-(90°)				[0°/90°]_2_				
	C-2D-(+45°)				[−45°/+45°]_2_				
	C-4D-(0°)	Carbon	4	Non-interlaced	[90°/±45°/0°]_1_	10	6400	10	15 × 15 ×15 × 15
	C-4D-(90°)				[0°/±45°/90°]_1_				
	C-4D-(+45°)				[+45°/90°/0°/−45°]_1_				
	C-4D-(−45°)				[+45°/0°/90°/−45°]_1_				

**Table 3 polymers-13-03073-t003:** Multiaxis 3D carbon concrete specifications.

Label	Panel Density (kg/m^3^)	Total Fiber Weight Fractions (%)	Partial Fiber Weight Fraction (%)	Void Content (%)	Water Absorption (%)
CC	2.190	-	-	22.080	11.160
C-1D-(0°)	2.190	2.540	2.540 (0°)	16.805	8.455
C-1D-(90°)	2.190	2.540	2.540 (90°)	16.805	8.455
C-1D-(+45°)	2.190	2.540	2.540 (+45°)	16.805	8.455
C-2D-(0°)	2.240	2.480	1.240 (0°), 1.240 (90°)	19.325	9.915
C-2D-(90°)	2.240	2.480	1.240 (90°), 1.240 (0°)	19.325	9.915
C-2D-(+45°)	2.240	2.480	1.240 (+45°), 1.240 (−45°)	19.325	9.915
C-4D-(0°)	2.170	2.560	0.640 (90°), 0.640 (−45°), 0.640 (+45°), 0.640 (0°)	19.675	9.960
C-4D-(90°)	2.170	2.560	0.640 (0°), 0.640 (+45°), 0.640 (−45°), 0.640 (90°)	19.675	9.960
C-4D-(+45°)	2.170	2.560	0.640 (+45°), 0.640 (90°), 0.640 (0°), 0.640 (−45°)	19.675	9.960
C-4D-(−45°)	2.170	2.560	0.640 (+45°), 0.640 (0°), 0.640 (90°), 0.640 (−45°)	19.675	9.960

**Table 4 polymers-13-03073-t004:** Average flexural test results on the multiaxis 3D carbon fiber concrete composites.

Label	Flexural First Crack Load (N)	Flexural First Crack Displacement (mm)	Flexural Max. Load (N)	Flexural Max. Displacement (mm)	Flexural First Crack Strength (MPa)	Flexural Max. Strength (MPa)	Flexural Stiffness (MPa/mm)
CC	1374.93 ± 190.83	0.29 ± 0.07	1374.93 ± 190.83	0.29 ± 0.07	6.19 ± 0.86	6.19 ± 0.86	27.33 ± 1.08
C-1D-(0°)	1234.72 ± 211.29	0.38 ± 0.07	3745.68 ± 355.63	4.35 ± 0.36	5.56 ± 0.95	16.87 ± 1.60	23.11 ± 2.97
C-1D-(90°)	920.16 ± 169.01	0.23 ± 0.04	920.16 ± 169.01	0.23 ± 0.04	4.14 ± 0.76	4.14 ± 0.76	22.56 ± 2.76
C-1D-(+45°)	1289.28 ± 219.06	0.36 ± 0.01	1289.28 ± 219.06	0.36 ± 0.01	5.81 ± 0.99	5.81 ± 0.99	25.49 ± 3.39
C-2D-(0°)	1473.87 ± 179.16	0.33 ± 0.06	2644.95 ± 348.85	3.11 ± 0.39	6.64 ± 0.81	11.91 ± 1.57	28.21 ± 3.72
C-2D-(90°)	1002.14 ± 110.65	0.28 ± 0.07	1988.02 ± 112.56	3.94 ± 0.41	4.51 ± 0.50	8.95 ± 0.51	22.12 ± 4.75
C-2D-(+45°)	1881.50 ± 219.61	0.52 ± 0.07	1881.50 ± 219.61	0.52 ± 0.07	8.47 ± 0.99	8.47 ± 0.99	27.95 ± 5.49
C-4D-(0°)	1134.04 ± 77.11	0.36 ± 0.05	2284.83 ± 209.79	2.93 ± 0.18	5.11 ± 0.35	10.29 ± 0.94	22.28 ± 2.18
C-4D-(90°)	1052.87 ± 107.59	0.31 ± 0.02	1052.87 ± 107.59	0.31 ± 0.02	4.74 ± 0.48	4.74 ± 0.48	23.84 ± 3.24
C-4D-(+45°)	1617.34 ± 446.58	0.63 ± 0.20	2799.47 ± 95.81	3.86 ± 0.46	7.28 ± 2.01	12.61 ± 0.43	20.41 ± 9.33
C-4D-(−45°)	1338.42 ± 606.06	0.42 ± 0.03	2137.16 ± 407.53	4.77 ± 1.01	6.03 ± 2.73	9.62 ± 1.84	21.22 ± 8.51

**Table 5 polymers-13-03073-t005:** Average off-axis flexural energy results for the multiaxis 3D carbon fiber concrete composites.

Label	Flexural First Crack Energy (J)	Flexural Panel Max. Load Energy (J)	Flexural Strain Hardening Energy (J)	Flexural Strain-Softening Energy (J)	Flexural Total Energy (J)
CC	0.16 ± 0.05	0.16 ± 0.05	0.00 ± 0.00	0.01 ± 0.00	0.17 ± 0.05
C-1D-(0°)	0.22 ± 0.08	11.23 ± 1.36	11.01 ± 0.72	7.84 ± 0.52	19.07 ± 2.39
C-1D-(90°)	0.09 ± 0.01	0.09 ± 0.01	0.00 ± 0.00	0.00 ± 0.00	0.09 ± 0.02
C-1D-(+45°)	0.16 ± 0.02	0.16 ± 0.02	0.00 ± 0.00	0.44 ± 0.07	0.60 ± 0.15
C-2D-(0°)	0.16 ± 0.04	6.19 ± 0.75	6.03 ± 0.40	6.83 ± 0.49	13.02 ± 1.73
C-2D-(90°)	0.10 ± 0.04	5.22 ± 0.63	5.12 ± 0.34	4.32 ± 1.23	9.54 ± 3.08
C-2D-(+45°)	0.26 ± 0.06	0.26 ± 0.06	0.00 ± 0.00	1.09 ± 0.32	1.35 ± 0.70
C-4D-(0°)	0.12 ± 0.02	4.00 ± 0.53	3.88 ± 0.28	2.34 ± 0.05	6.34 ± 0.63
C-4D-(90°)	0.10 ± 0.01	0.11 ± 0.81	0.01 ± 0.41	6.77 ± 0.13	6.88 ± 0.56
C-4D-(+45°)	0.28 ± 0.09	7.01 ± 0.53	6.73 ± 0.31	3.54 ± 0.21	10.55 ± 0.95
C-4D-(−45°)	0.17 ± 0.08	6.75 ± 0.10	6.58 ± 0.09	3.77 ± 0.25	10.52 ± 0.60

**Table 6 polymers-13-03073-t006:** Off-axis flexural failure results of multiaxis 3D carbon fiber concrete.

Label	Number of Cracks		Crack Length (mm)		Crack Width (mm)		Crack Spacing (mm)
	Top Side	Bottom Side	Top Side	Bottom Side	Top Side	Bottom Side	Bottom Side
CC	Catastrophic failure	Catastrophic failure	-	75	-	-	-
C-1D-(0°)	-	2	-	75	-	0.585	103.80
C-1D-(90°)	Catastrophic failure	Catastrophic failure	-	75	-	-	-
C-1D-(+45°)	-	1	-	75	-	0.217	-
C-2D-(0°)	-	2	-	75	-	0.630	88.60
C-2D-(90°)	-	2	-	75	-	0.602	68.90
C-2D-(+45°)	-	1	-	75	-	0.216	-
C-4D-(0°)	-	3	-	75	-	0.612	52.40–68.10
C-4D-(90°)	-	1	-	75	-	2.102	-
C-4D-(+45°)	-	2	-	75	-	0.461	37.30–73.90
C-4D-(−45°)	-	3	-	75	-	0.685	53.30–78.10

## Data Availability

Data is provided in the article.

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
