# Peer review of "Experimental Study on Angular Flexural Performance of Multiaxis Three Dimensional (3D) Polymeric Carbon Fiber/Cementitious Concretes"

_polymers, 2021, doi:10.3390/polym13183073_

Round 1

Reviewer 1 Report

The manuscript is within the scope of polymers and it should be published, however before some major concerns must be addressed, namely:

1- The introduction is extremely complete. However, I beleive that it would benefit in reducing the number of citations since the author cited 88 references and some could be avoided.

2 - Figures 6, 7, 8, 9, 10 and 14 must be represented in a way that when printed in black and white it is easy to be distinguished between each other. As presented it is not possible.

3 - The quality of figures 11, 12, 15, 18 and 19 must be improved.

4 -Conclusions need to be shortened and go straight to the point.

Author Response

RESPONSE TO REVIEWER(S)’ COMMENTS [Manuscript ID: Polymers-1362104 ]

 Date: 4 September 2021

Reviewer comments:

Reviewer 1

Comments and Suggestions for Authors

The manuscript is within the scope of polymers and it should be published,

Answer: Thank you for the comment and decision.

however before some major concerns must be addressed, namely:

1- The introduction is extremely complete. However, I beleive that it would benefit in reducing the number of citations since the author cited 88 references and some could be avoided.

Answer: The required revision on the manuscript was made. Right now, reference in the manuscript was reduced from 107 to 100, especially self-citation was shortened from 13 to 5.  

2 - Figures 6, 7, 8, 9, 10 and 14 must be represented in a way that when printed in black and white it is easy to be distinguished between each other. As presented it is not possible.

Answer: The required revision on the mentioned Figures 6, 7, 8, 9, 10 and 14 were revised to exhibit more clearly. On the other hand, the color effects we used in the Figures were probably shown based on color tone differences.    

3 - The quality of figures 11, 12, 15, 18 and 19 must be improved.

Answer: The quality of the Figures 11, 12, 15, 18 and 19 were improved especially using Adobe Photoshop, and resolutions were improved from 96 to 300 dpi. In addition, all Figures resolutions were checked and some of them were changed with high (300) dpi resolution figures and figure captions were highlighted red color.

4 -Conclusions need to be shortened and go straight to the point.

Answer: The required revision on the conclusion was made and it was shortened and highlighted red color.

We think that the required revisions on the manuscript based on reviewer comments were nicely completed and we would like to do additional revision, if required.

My best regards,

Reviewer 2 Report

Figure 1 - please change the colors of text, because it is invisible.

Please explain TOW, because no explanation is provided. I am aware of the meaning of the word tow, but using capital letters suggest some abbreviation.

Figure 2 - please change the colors of text, because it is invisible.

Was the curing time previously optimized? Because maybe 21 days would be enough?

Figure 4 c-f makes no sense, because differences are practically invisible.

"Thus, almost flawless multiaxis 3D carbon concrete composite samples were made for the flexural test" - please explain how samples with over 16% void content are almost flawless. Refer to the typical values for concrete composites, because for polymers such a high value is unacceptable and the issue could not be clear for readers.

Same for water absorption. Discuss presented results, refer to the literature. For now, there are only values provided in Table without discussion and analysis.

The results of flexural tests are nicely presented from multiple angles, but the results description lacks scientific discussion and refering to other literature reports. Currently it is rather report from analysis than research paper. Part about flexural tests, void content and water absorption does not contain any reference, which is unacceptable in research paper.

About the FTIR analysis, what was the goal? Authors only changed the allignment of fibers. What changes in FTIR spectra Authors wanted to see? Any chemical reactions? If so, please provide schemes.

"From the FTIR analysis, it was identified comparatively that a strong physical interaction between the cementitious components and PAN carbon fiber (filament TOWs) was realized" - how did Authors see the strong physical interactions? Please point to the particular peaks in the FTIR spectra.

In the FTIR spectra why the peaks for carbon fibers are reversed?

What is the reason for the differences in FTIR spectra of warp-cross sections for C-1D and C-4D samples, pleaso refer to the particular peaks and provide explanation based on the chemical structure of both samples.

Failure analysis looks noticeably better than flexural tests analysis, however, still lacks the refering to literature.

Author Response

RESPONSE TO REVIEWER(S)’ COMMENTS [Manuscript ID: Polymers-1362104 ]

Date: 4 September 2021

Reviewer comments:

Reviwer 2.

Comments and Suggestions for Authors

Figure 1 - please change the colors of text, because it is invisible.

Answer: The required revision on the Figure 1 was made in order to clearly visible for readers and Figure caption highlighted red color.   

Please explain TOW, because no explanation is provided. I am aware of the meaning of the word tow, but using capital letters suggest some abbreviation.

Answer: The TOW is not abbreviations in textile science; rather it is definition as “an untwisted bundle of continuous filaments”. It was defined in the first instance related part of the manuscript which is introduction part and highlighted red color.

Figure 2 - please change the colors of text, because it is invisible.

Answer: The required revision on the Figure 2 was made in order to clearly visible for readers and Figure caption highlighted red color.   

Was the curing time previously optimized? Because maybe 21 days would be enough?

Answer: We generally followed the literature and some collaborative support from department of Civil engineering both in Erciyes University and Gaziantep University. After long critical discussion for curing time (7 days, 14 days, 21 days, 28 days, 3 months or 6 months), we decided to consider the 28 days for curing the carbon fiber concrete panel for flexure test partly due to fiber and cementitious bonding. Even, we dipped the filament TOW in the cementitious matrix and observed the outcome at certain some time period (around a month). Fiber and cement matrix were almost perfectly bonded as seen in SEM images (Figure 5).  However, for future, more experimental studies must be carried out to identify the best bonding conditions via functionalization of fiber or grafting between cement matrix and fiber.

Figure 4 c-f makes no sense, because differences are practically invisible.

Answer: The required revisions on the Figure 4 c-f were made and became visible for reader stand points, and Figure 4 (a-f) caption was highlighted red color.

"Thus, almost flawless multiaxis 3D carbon concrete composite samples were made for the flexural test" - please explain how samples with over 16% void content are almost flawless. Refer to the typical values for concrete composites, because for polymers such a high value is unacceptable and the issue could not be clear for readers.

Answer:  Referee 2 is right based on polymer material properties concerns. The required revision was made on the related part of the paragraph in the results and discussion under the sub title of “3.1. Fiber volume fraction and density results.” The high void percent in the concrete structure is another issue to discuss for maybe separate paper due to complex reactions were occurred during curing in matrix and somehow between filament TOW and cement matrix. This is extremely complicated issues which require interdisciplinary actions for future research endeavor. On the other hand, we are researching thermal, electrical, sound insulation and air permeability properties of the continuous fiber based concretes to understand the effect of void, functionalization of fiber and nano additions etc. We would like to disseminate the finding with readers soon.

Same for water absorption. Discuss presented results, refer to the literature. For now, there are only values provided in Table without discussion and analysis.

Answer: Reviewer 2 is right. However, we did not discussed or pay attention for water absorption or void contents. This is because these properties will be integrated with thermal, electrical, sound insulation and air permeability properties of the continuous fiber based concretes to understand the effect of void or water absorption properties at the current research study, and the finding will be published for near future.  

The results of flexural tests are nicely presented from multiple angles, but the results description lacks scientific discussion and refering to other literature reports.

Answer: I am afraid disagree with referee 2. The results were well discussed considering reason-result relations especially for some critical parameter stand points via using nicely proper figures and Tables. We developed almost unique multiaxis fiber based concrete structure. Their flexural load-displacement curves, flexural strength and flexural energy were calculated based on using some useful tools such as MATLAB and EXEL. These results were compared to each other and certain critical findings were discovered following the scientific principles via generated data.    

Currently it is rather report from analysis than research paper. Part about flexural tests, void content and water absorption does not contain any reference, which is unacceptable in research paper.

Answer: I am little bit disagree with Referee 2 due to our objective is not research the effect of void content and water absorption on the flexural strength properties of the multiaxis three dimensional continuous carbon fiber cementitious concrete. Instead, our objective is to find out the effect of fiber direction to the flexural properties of developed concrete structure. Therefore, this finding was well discussed and well presented in the manuscript. However, we will do further research on the effect of void and water absorption to the thermal, electrical, sound insulation and air permeability properties of the continuous fiber based concretes. We are currently working on it. When they are ready, we would like to disseminate the finding with readers.

About the FTIR analysis, what was the goal?

Answer: The goal of the FTIR is to search the relations between the PAN based continuous carbon fiber and cementitious matrix especially interphase regions to understand the bonding phenomena.   

Authors only changed the allignment of fibers. What changes in FTIR spectra Authors wanted to see?

Answer: This is somehow true that we changed the fiber alignment and this led to the change of the directional amount of fiber fractions (wt. %). We would like to see the effect of the fiber orientation on the flexural strength, energy absorption capacity of the multiaxis 3D continuous carbon/cementitious concrete structure. Additionally, we checked the perfectness between cementitious matrix and fiber boundaries for bonding stand points. Thus, we measured the FTIR of developed structure to understand the iterations between fiber and cementitious matrix. After this experience, we get the idea that fiber coating or surface modification of carbon fiber via suitable coating method can be appropriate to get better bonding. This will be one of the future research priorities for study.     

Any chemical reactions? If so, please provide schemes.

Answer: Probably physical reactions rather than chemical reaction due to matrix reactions based on electrochemical properties. This can be considered that there is a complex chemical reactions between cement sub component and sand, water and superplasticizer around the continuous carbon fiber. Thus, carbon fiber was locked in its place surrounding cementitious matrix. In addition, we indicated in the manuscript that the fiber placement and orientation in the concrete did not affect their FTIR spectrum.

"From the FTIR analysis, it was identified comparatively that a strong physical interaction between the cementitious components and PAN carbon fiber (filament TOWs) was realized" - how did Authors see the strong physical interactions? Please point to the particular peaks in the FTIR spectra.

Answer: We did not identified chemical reaction based on FTIR spectrum. However, we identified from the FTIR analysis that a strong physical interaction between the cementitious components and PAN carbon fiber (filament TOWs) was realized based on the SEM images (Fig. 22 and Fig. 23). This also indicated that their directional flexural properties were homogeneously distributed throughout the concrete structure. As a result, the C-4D structure controlled the number of cracks, crack width, crack distance, and improved the flexural energy absorption capacity of the concrete structure. However, more study is required, especially regarding the cement-fiber interlaminar regions in the multiaxis 3D carbon fiber concrete composites. We also did some minor revision in the FTIR part of the manuscript and highlighted red color.

Therefore, we would like to demonstrate that from flexural test, somehow FTIR and some images, the flexural properties were probably uniformly distributed along the fiber orientations. In addition, we made the flexural samples uniform, means, cementitious matrices are homogenously distributed around the directional filament TOWs.     

In the FTIR spectra why the peaks for carbon fibers are reversed?

Answer: We used Mid-Infrared Spectroscopy to obtain FTIR spectrum. The wave length (cm-1) versus Transmittance (%) spectrum is between 4000 and 500 cm-1. Generally, open literature shows the FTIR spectra peaks from 4000 to 500 cm-1. So, we present the FTIR spectra from 4000 to 500 cm-1 where I think that we have a right as an author(s). The carbon fiber FTIR spectrum obtained from Mid-Infrared Spectroscopy is shown in Figure 1.   

Please note that Figure 1 a and b was not shown in this window for some reason, but it is shown in PDF form of the reviewer responce.

Figure 1. (a) PAN carbon fiber FTIR spectrum (top) and (b) condense form of PAN carbon fiber FTIR spectrum via Exel (bottom).  

What is the reason for the differences in FTIR spectra of warp-cross sections for C-1D and C-4D samples, pleaso refer to the particular peaks and provide explanation based on the chemical structure of both samples.

Answer: The FTIR spectra obtained from C-1D and C-4D samples were well defined and discussed in the related part of the manuscript. In the FTIR spectrum of the C1-D (warp cross-section), C-2D (filling cross-section), and C-4D (warp cross-section) PAN carbon/cementitious concrete composites (Fig. 16), a common signal on all developed structures were detected. The peak observed between 3500 cm-1 and 3000 cm-1 bands may be attributed to the deformation vibrations of H2O, polycarboxylate, -Si-OH, and -Ca-OH groups, In the 3000-2850 cm-1 band spacing, the deformation vibrations of -CH2-CH- (aliphatic) groups were observed, whereas the signal between 2300 cm-1 and 2200 cm-1 corresponded to the −C≡N (nitryl) group.  The signals between 1600 cm-1 and 1300 cm-1 were attributed to stretching vibrations of the -C=C-, -C=O, and -C=N groups. The signal observed between 1100 and 1000 cm-1 was assigned to Si-O asymmetric stretching vibrations and another signal in the 900-500 cm-1 band spacing was considered the stretching vibrations of single bond groups.

In the FTIR spectrum of C-4D (-bias cross-section), C-4D (+bias cross-section), C-4D (filling cross-section), and C-4D (warp cross-section) as exhibited in Fig. 17, almost similar results were obtained compared to the C-1D (warp cross-section) and C-2D (filling cross-section) (Fig. 16). It was considered that the fiber placement and orientation in the concrete did not affect their FTIR spectrum. From the FTIR analysis, it was identified comparatively that probably a strong physical interaction between the cementitious components and PAN carbon fiber (filament TOWs) was realized. These findings were supported by fractured sample SEM images (Fig. 22 and Fig. 23). Hence, their directional flexural properties were homogeneously distributed throughout the concrete structure. As a result, the C-4D structure controlled the number of cracks, crack width, crack distance, and improved the flexural energy absorption capacity of the concrete structure. However, more study is required, especially regarding the cement-fiber interlaminar regions in the multiaxis 3D carbon fiber concrete composites.

Failure analysis looks noticeably better than flexural tests analysis, however, still lacks the refering to literature.

Answer: First of all, the flexural test analysis were nicely performed based on systematic step by step and vis-à-vis comparison between developed structures as C-1D, C-2D and C-4D and their angular cases. Multiaxis three dimensional continuous carbon fiber /cementitious concrete structures are a new type of concrete structure and it is hard to compared to the exact one in the open literature. That is why the comparisons were made mostly between developed structure types as C-1D, C-2D and C-4D and their angular cases. Same is true for failure analysis case.

We think that the required revisions on the manuscript based on reviewer comments were nicely completed and we would like to do additional revision, if required.

My best regards,

Reviewer 3 Report

There are some weaknesses through the manuscript which need improvement. Therefore, the submitted manuscript cannot be accepted for publication in this form, but it has a chance of acceptance after a minor revision. My comments and suggestions are as follows:

1- Abstract gives information on the main feature of the performed study, but some details about the obtained results must be added.

2- Authors must clarify necessity of the performed research. Objectives of the study, must be clearly mentioned in introduction.

3- The literature study must be enriched. In this respect, authors must read and refer to the following papers: (a) https://doi.org/10.1007/s10921-020-00721-1 (b) https://doi.org/10.1016/j.ijpharm.2020.119732

4- Why these particular layer and configurations were selected.

5- It is necessary to put scale bar in the figures, for instance in figure 1.

6- Details of calculations (values presented in Table 4) must be added.

7- Deviation (in presented curves, figure 8) and error in calculations must be discussed.

8- In its language layer, the manuscript should be considered for English language editing. There are sentences which have to be rewritten.

9- The conclusion must be more than just a summary of the manuscript. List of references must be updated based on the proposed papers. Please provide all changes by red color in the revised version.

Author Response

RESPONSE TO REVIEWER(S)’ COMMENTS [Manuscript ID: Polymers-1362104 ]

 Date: 4 September 2021

Reviewer comments:

Reviewer 3.

Comments and Suggestions for Authors

There are some weaknesses through the manuscript which need improvement. Therefore, the submitted manuscript cannot be accepted for publication in this form, but it has a chance of acceptance after a minor revision.

Answer: We can do required revisions on the manuscript.

My comments and suggestions are as follows:

1- Abstract gives information on the main feature of the performed study, but some details about the obtained results must be added.

Answer: Some detail explanations were added in the abstract and highlighted red color.

2- Authors must clarify necessity of the performed research. Objectives of the study, must be clearly mentioned in introduction.

Answer: The necessity of the research in the manuscript was explained in the introduction part of the manuscript and highlighted red color.

3- The literature study must be enriched. In this respect, authors must read and refer to the following papers: (a) https://doi.org/10.1007/s10921-020-00721-1 (b) https://doi.org/10.1016/j.ijpharm.2020.119732

Answer: The above mentioned literatures were included in the introduction part of the manuscript and highlighted red color.

4- Why these particular layer and configurations were selected.

Answer: We developed three types of structures as C-1D, C-2D and C-4D due to identify the directional effects of fiber on the flexural properties including strength and energy absorptions. These structures represent axial (C-1D), biaxial (C-2D) and multiaxis (C-4D) directional concretes and these directional arrangements influence the flexural strength and flexural energy absorption properties. Therefore, depending on the load-carrying requirement and matrix types, these can be applicable some application in the civil engineering even any other engineering field as well. I have almost 35 years’ experience on the multiaxis preform, fiber formations in the subject matter. Thus, the selection was based on experience, intuitions, engineering principle and design as well as free style creativity inspired from nature.     

5- It is necessary to put scale bar in the figures, for instance in figure 1.

Answer: We used a digital camera (Nikon D3000 10.2MP Digital SLR Camera with 18-55mm f/3.5-5.6G AF-S DX VR Nikkor Zoom Lens, JP). We did not use the optical camera and of course not a SEM to use the scale bar. Therefore, this is not applicable.

6- Details of calculations (values presented in Table 4) must be added.

Answer: Details of the calculations were already provided throughout the manuscript. For instance, Equations (1) and (2) were provided in the material and method section of the manuscript; Calculation was also conducted by using MATLAB R2016a (The MathWorks, Inc., USA). This was accomplished by applying the numerical integration and standard plotting tools of MATLAB [93]. After the flexural test data for the pristine (CC) and carbon concrete (C-1D-(0°), C-1D-(90°), C-1D-(+45°), C-2D-(0°), C-2D-(90°), C-2D-(+45°), C-4D-(0°), C-4D-(90°), C-4D-(+45°) and C-4D-(-45°)) obtained from the Shimadzu AG-X 100 (JP) testing machine, they were transferred to the Microsoft (MS) Excel spreadsheet 2013 via Trapezium® software. Therefore, statistical computations on the raw data including flexural first crack load and displacement, flexural maximum load and displacement, flexural first crack strength, and flexural maximum strength, as well as flexural stiffness were carried out and analyzed (as presented in Table 4).

7- Deviation (in presented curves, figure 8) and error in calculations must be discussed.

Answer: The deviation in calculations at the manuscript was discussed briefly and highlighted red color.

8- In its language layer, the manuscript should be considered for English language editing. There are sentences which have to be rewritten.

Answer: The manuscript was again checked for the English language by me and our department staff, and required corrections were made and highlighted red color. 

9- The conclusion must be more than just a summary of the manuscript.

Answer: The required revision on the conclusion of manuscript was made and highlighted red color.

 List of references must be updated based on the proposed papers.

Answer: The reference list in the manuscript was revised and highlighted red color.

Please provide all changes by red color in the revised version.

Answer: All changes and revision in the manuscript were highlighted red color.

We think that the required revisions on the manuscript based on reviewer comments were nicely completed and we would like to do additional revision, if required.

My best regards,

Round 2

Reviewer 1 Report

I beleive the authors did not understand what I mean when mentioned that Figures 6, 7, 8, 10 and 14 should be represented in a way that when printed in black and white it is possible to distinguish between each sample. Use patterns (dashed lines or points) or identify each plot such as is presented in Figure 16.

Author Response

RESPONSE TO REVIEWER(S)’ COMMENTS [Manuscript ID: Polymers-1362104 ]

 Date: 6 September 2021

Reviewer comments:

 Reviewer 1

Comments and Suggestions for Authors

I beleive the authors did not understand what I mean when mentioned that Figures 6, 7, 8, 10 and 14 should be represented in a way that when printed in black and white it is possible to distinguish between each sample. Use patterns (dashed lines or points) or identify each plot such as is presented in Figure 16.

Answer: The required revision on the mentioned Figures 6, 7, 8, 9, 10 and 14 were revised to distinguish between each sample. For this reason, we used various forms of patterns via excel spreadsheet and Figure captions were highligted red color.

My best regards,

Reviewer 2 Report

Everything in order after corrections

Author Response

RESPONSE TO REVIEWER(S)’ COMMENTS [Manuscript ID: Polymers-1362104 ]

 Date: 6 September 2021

Reviewer comments:

Reviwer 2.

Comments and Suggestions for Authors

Everything in order after corrections.

Answer: Thank you for the decision.

My best regards,

Reviewer 3 Report

The paper has been improved and some modifications have been conducted. I think it can be considered for publication.

Author Response

RESPONSE TO REVIEWER(S)’ COMMENTS [Manuscript ID: Polymers-1362104 ]

 Date: 6 September 2021

Reviewer comments:

Reviewer 3.

Comments and Suggestions for Authors

The paper has been improved and some modifications have been conducted.

Answer: Thank you for the comment.

I think it can be considered for publication.

Answer: Thank you for the decision.

My best regards,